# A Plug-and-Play Agentic Framework for Text Guided Image Editing

## Abstract

Text-guided image editing with diffusion models struggles to maintain fidelity during complex, multi-aspect edits, where simultaneous changes and preservations are required. While one-shot prompt rewriting offers some improvement, it lacks fine-grained control, often leading to under-editing of desired attributes or over-editing of unrelated regions, or both. To address these gaps, we introduce an agent called `ARTIE` (**A**uditable **R**efinement for **T**ext-Guided **I**mage **E**diting), which is a plug-and-play, inference-time, feedback-based agentic system to enhance pre-trained diffusion models such as Stable Diffusion. At its core, `ARTIE` is organized into three agentic sub-modules: (1) a perception module (`SceneDiff`), which detects over-editing by comparing source and target scene graphs, and under-editing by grounding edit requirements through a dual-verification pipeline comprising an open-set object detector and CLIP; (2) a reasoning/planner module (an LLM-based Prompt Engineer), which takes the diagnostic signals from `SceneDiff` and synthesizes refined positive prompts together with asymmetrically weighted negative prompts; and (3) an action module (the image generator), which executes these refined prompts to produce improved images iteratively. This perception–reasoning–action loop runs in multiple cycles, producing high-quality edited images. Consequently, `ARTIE` also yields an auditable trail of refinement steps where each modification is explained and justified via explicit feedback signals from the perception module. Further, `ARTIE` operates solely through guided prompt engineering, without requiring model retraining or fine-tuning, making it a plug-and-play architecture. Despite being training-free, when applied on top of Stable Diffusion, `ARTIE` consistently improves fidelity and control in multi-aspect editing. Its performance matches or surpasses specialized baselines, thereby setting a new state-of-the-art for explainable, inference-time agentic image editing.

## 1 Introduction

Image editing encompasses a range of tasks, from pixel adjustments (e.g., brightness, contrast) to high-level changes (e.g., object addition, style transfer). Editing methods range from traditional manual tools (Photoshop, GIMP) and classical algorithmic approaches (histogram equalization, seam carving), to modern generative techniques (GANs, VAEs, diffusion models) for realistic, data-driven edits. Recently, text-guided generative editing has emerged as a flexible and intuitive method, enabling users to describe edits using free-form language.

GAN and VQ-based approaches (Patashnik et al., 2021; Xia et al., 2021; Li et al., 2020) linked text to latent manipulations but were limited in domain coverage and edit locality. Diffusion models now dominate due to their realism and controllability, enabling a range of methods: mask-based inpainting (Nichol et al., 2021; Lugmayr et al., 2022), attention steering (Hertz et al., 2022; Cao et al., 2023), inversion for real-image fidelity (Mokady et al., 2022; Kawar et al., 2023), and instruction-following editors (Brooks et al., 2023). Together, these techniques support a wide range of edits across real and synthetic images. Despite progress, two recurring challenges persist – (1) Locality: ensuring edits occur only where intended, without distorting unrelated content. (2) Consistency: ensuring edits match the textual description while preserving non-edited regions (Qian et al., 2025). These shortcomings often manifest as under-editing (missed changes) or over-editing (unintended alterations), sometimes both simultaneously (Kim et al., 2025). The burden often falls on the user to carefully craft prompts or supply auxiliary inputs, which limits reliability in real-world scenarios.

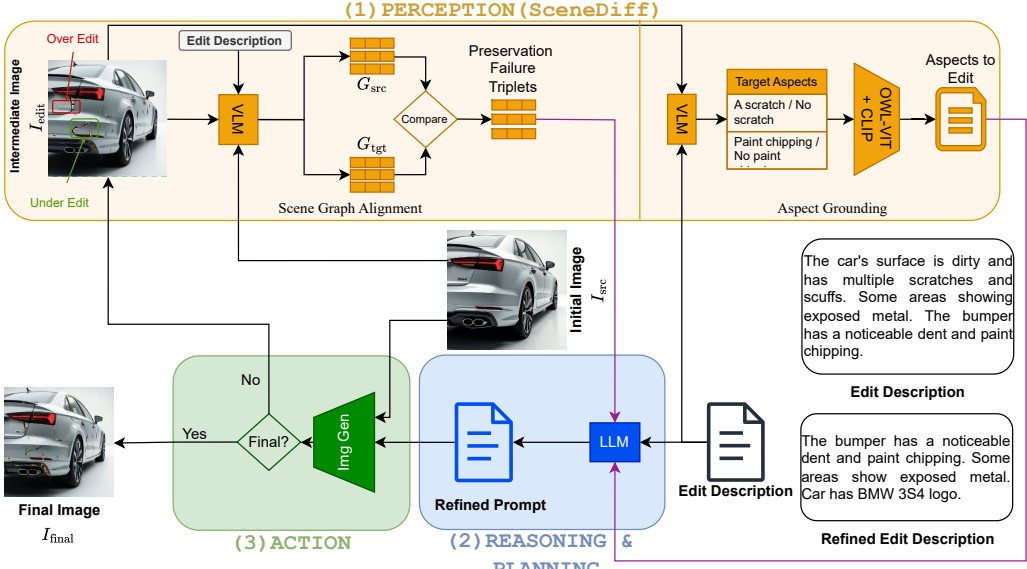

Figure 1: *Simplified* overview of the `ARTIE` pipeline for text-guided image editing. The `SceneDiff` perception module diagnoses faulty edits (initially generated through the action module) through scene graph alignment and aspect grounding, providing structured feedback to an LLM-based reasoning and planning module for prompt refinement. The action module executes these refined prompts to produce improved edits. Once the initial edited image is available, the perception module comes into play from the 2nd iteration onwards (illustrated through magenta arrows).

To overcome these challenges, we present `ARTIE` (**A**udit-trail based **R**efinement for **T**ext-Guided **I**mage **E**diting), an agentic framework designed as a **plug-and-play** wrapper for pre-trained models like Stable Diffusion. As a purely **inference-time** method, it introduces a feedback loop for faithful, controllable, multi-aspect editing. The agent's *perception module*, called `SceneDiff`, is its core verifier that detects over-editing through scene graph alignment and under-editing via aspect-level grounding. These structured signals are fed into its *reasoning & planning module*, an LLM-based Prompt Engineer, which generates refined positive prompts and asymmetrically weighted negative prompts to drive a multi-cycle correction loop. Unlike prior art, `ARTIE` requires no retraining or fine-tuning, and produces an audit trail of corrections, making it both effective and explainable. Our key contributions and the novelty of the solution are as follows:

- **Agentic framework for text-guided image editing:** An inference-time multi-cycle pipeline that combines `SceneDiff`-based verification, using scene graphs and aspect-level grounding to detect over/under-editing, with LLM-driven refinement via automated prompt engineering, employing refined positive and asymmetrically weighted negative prompts for accurate editing.

- **Built-in explainability:** Disentanglement of preservation and editable aspects into an explicit set of concepts keeps an audit of the progress and makes the agentic loop explainable.

- **Competitive performance without fine-tuning:** Demonstrates significant fidelity gains over base diffusion models and naive prompt engineering, achieving performance that is competitive with, and often exceeds, specialized fine-tuned models on multi-aspect edits.

## 2 RELATED WORK

Learning-based editing leverages generative models (GANs, VAEs, diffusion) to produce realistic, semantic-level modifications, guided by signals such as latent directions, sketches, bounding boxes, or increasingly, natural language descriptions.

Among control modalities, text has become the most natural and flexible. Early methods integrated CLIP (Radford et al., 2021a) with GANs, VAEs, or VQ models to tie prompts to latent manipulations.

GAN-based editors such as StyleCLIP (Patashnik et al., 2021), TediGAN (Xia et al., 2021), and ManiGAN (Li et al., 2020) enabled attribute-level edits (e.g., "add a smile") and domain adaptation (Gal et al., 2021). While pioneering, these approaches were limited by domain dependence, inversion artifacts, and coarse edits—challenges that motivated the shift to diffusion-based models.

Diffusion models dominate text-guided editing due to their realism, semantic coverage, and controllability. Mask-based inpainting methods (Nichol et al., 2021; Lugmayr et al., 2022; Meng et al., 2022) provide strong locality via explicit masks but require user input. To reduce reliance on explicit spatial supervision, mask-free editing approaches (Couairon et al., 2023) infer editable regions automatically, though sensitive to prompt phrasing. Another important strand of work (Hertz et al., 2022; Cao et al., 2023; Parmar et al., 2023) focuses on preserving layout and identity by reusing attention maps, enabling training-free, structure-preserving edits. Inversion methods (Mokady et al., 2022; Kawar et al., 2023; Huberman-Spiegelglas et al., 2024) are central for faithful edits of real images, though often requiring per-image optimization. Brooks et al. (2023) allows natural-language instructions in a single pass, but relies on synthetic training data and is less reliable for identity-sensitive edits. Multimodal grounding (Zhang et al., 2023; Li et al., 2023) enforces geometry or spatial constraints via edges, depth maps, poses, or bounding boxes, offering strong controllability but at the expense of requiring auxiliary annotations. Layered/compositional approaches (Bar-Tal et al., 2022; Yang et al., 2022) predict overlay layers or use exemplar guidance for localized, non-destructive edits.

Early LLM-guided diffusion systems, including self-correcting LLM-controlled Diffusion (Wu et al., 2023) and LLM-grounded Diffusion (Lian et al., 2024), focus primarily on text-to-image generation. These methods employ open-loop pipelines or latent-space manipulations for generation or refinement, and rely on ad-hoc error-handling strategies such as external modules or one-shot verification. Building on these foundations, emerging agentic frameworks replace single-pass editing with a perception–reasoning–action loop, where verification modules diagnose flaws and specialised tools iteratively refine images. Representative systems include *GenArtist* (Wang et al., 2024a) coupling multimodal LLMs with diverse backends, *RefineEdit-Agent* (Liang et al., 2025) enabling training-free multi-turn edits, *RPG* (Yang et al., 2024) introducing region-wise planning and diffusion, and *CompAgent* (Wang et al., 2024b) employing a divide-and-conquer strategy for complex prompts. These approaches advance fidelity, control, and interpretability through structured iterative feedback.

Despite these advances, two persistent challenges remain: (i) locality—ensuring edits affect only the intended regions without distorting unrelated content; and (ii) consistency—ensuring edits align with textual instructions while preserving layout and identity. Existing methods address these issues only partially, often shifting the burden to users, who must craft precise prompts, provide masks, or manually select tools. These limitations underscore the need for an agentic workflow that can automatically interpret instructions, orchestrate editing mechanisms, and evaluate results. By closing the loop between editing, verification, and correction, such systems promise more reliable and accessible text-guided editing. However, none of the existing approaches explicitly handle multi-aspect fidelity at inference time, a gap our proposed method, ARTIE, is designed to fill. Our method, ARTIE, introduces three key innovations: (i) a disentangled verification pipeline that balances over-editing and under-editing; (ii) an explicit audit trail that enhances explainability; and (iii) a prompt-editing mechanism that improves the fidelity of existing image generation model, without retraining or relying solely on external tools.

## 3 METHODOLOGY: THE ARTIE AGENTIC LOOP

We frame multi-aspect image editing not as a single-pass generation problem, but as an iterative reasoning task performed by our agent, ARTIE. The agent's workflow is designed to mimic a human behavior: it first **generates** a candidate edited image, **inspects or perceives** it for flaws, **reasons** about the cause of those flaws, bakes a refinement prompt to mitigate these flaws (**planning**), and finally **acts** by generating an improved image. This *perception-reasoning/planning-action* cycle repeats until the edit is satisfactory. Figure 1 illustrates all the building blocks of ARTIE and in what follows we describe each of them.

## 3.1 Perception: Inspecting the Edit with SceneDiff

The first step in any correction loop is to identify the flaws. The agent needs to answer two critical questions: (i) Did the underlying image editing model fail to make a requested change (an *under-edit*)? and (ii) Did the model induce an unwanted alteration (an *over-edit*)?

In ARTIE, this perception block is called as SceneDiff module. This module is a dual-pipeline verifier that begins by taking as input the source image $I_{src}$, an intermediate edited image $I_{edit}$, and an edit description $D_{tgt}$. Next, it systematically inspects $I_{edit}$ from two complementary perspectives.

### 3.1.1 Detecting Over-Editing via Scene Graph Alignment

To detect unwanted alterations (*aka* over-edits) in $I_{edit}$, the agent first constructs scene graphs from both $I_{src}$ and $I_{edit}$ images. Next, it compares these scene graphs to form preservation failure triples $p_i$ as shown in Figure 2 (and 8) as well as explained below.

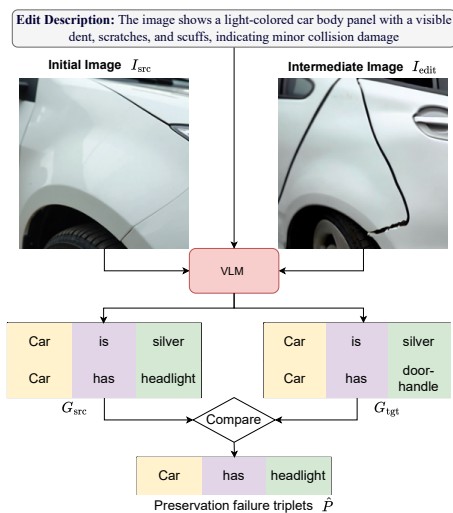

Figure 2: An overview of preservation analysis in our agentic pipeline. The VLM is provided with two separate in-context prompts for source and target scene graph construction (not shown in the figure). The VLM module generates both source and target scene graphs.

**Scene graphs construction:** Using a Vision-Language Model (VLM), LLaMA-4 Maverick, the agent analyses $I_{src}$ and $D_{tgt}$ to construct a *preservation-centric scene graph*, $G_{src}$. The VLM is prompted to identify all objects, attributes, and relations in $I_{src}$ that are *not* mentioned in the edit description $D_{tgt}$. These constitute the 'edit-invariant aspects' of the scene, which are the aspects of the scene that should remain unchanged while editing. Next, the agent creates a scene graph, $G_{tgt}$, for the edited image $I_{edit}$ using the same VLM.

**Identification of preservation failures triplets ($p_i$):** Taking a set difference, $\hat{P} = G_{src} \setminus G_{tgt}$, we produce a set of *preservation failures*. Each failure is a triplet, $p_i$ from the original scene graph that was unintentionally lost or altered during the edit, providing a precise, structured signal for an over-edited feature.

### 3.1.2 Detecting Under-Editing via Aspect Grounding

Now the agent verifies whether the intended edits from $D_{tgt}$ were successfully rendered or some edits got missed. Figure 3 (and 9) illustrates the process of identifying missing edits. This process works as follows:

**Gathering edit requirements ($q_i$) from $D_{tgt}$:** The VLM first decomposes the free-form description $D_{tgt}$ into a set of atomic, verifiable requirements, $Q = \{q_1, q_2, ...\}$. For each requirement $q_i$, a negated counterpart $\hat{q}_i$ is also generated (*e.g.*, if $q_i$ is 'a blue car,' $\hat{q}_i$ is 'not a blue car').

**Detecting edit requirements failures:** Each requirement $q_i$ is then checked against the intermediate edited image $I_{edit}$ using a two-stage pipeline comprising of OWL-ViT (Minderer et al., 2022) and CLIP (Radford et al., 2021b) models. The OWL-ViT is an open-set detector. It takes an image $I_{edit}$ and a requirement $q_i$ as input and returns a bounding box $b_i$ in the $I_{edit}$ image, highlighting the possible region of $I_{edit}$ corresponding to $q_i$. Next, this bounding box, along with $q_i$ and $\hat{q}_i$, is fed to the CLIP model. The CLIP model first embeds $q_i$ and $\hat{q}_i$ into $t(q_i)$ and $t(\hat{q}_i)$ vector embeddings, respectively, using a text encoder $t(\cdot)$. Next, it also embeds the bounding box region $b_i$ into a vector embedding $v(b_i)$ using the image encoder $v(\cdot)$. Now, we compute cosine similarity between these embeddings: $sim(b_i, q_i) = \frac{v(b_i) \cdot t(q_i)}{||v(b_i)|| \cdot ||t(q_i)||}$ and similarly, for $sim(b_i, \hat{q}_i)$. If the cosine similarity is higher for the negated requirement, that is $sim(b_i, \hat{q}_i) > sim(b_i, q_i)$, we add it to the set of under-editing failures, denoted by $\hat{Q}$. For an in-depth discussion of this module, please refer to Section A of the Appendix.

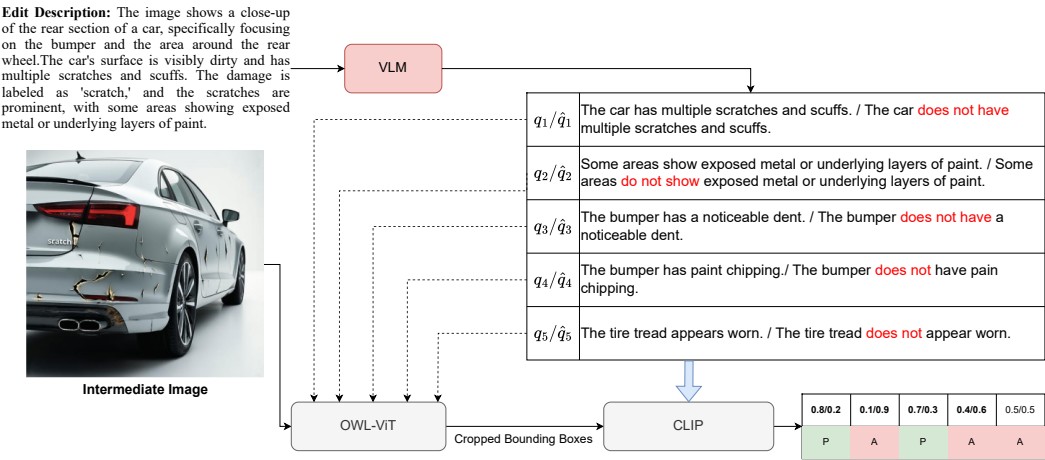

Figure 3: Detecting under editing with aspect grounding. A VLM extracts target aspects from the text prompt. An OWL-ViT+CLIP system then verifies each aspect against the edited image. **P** and **A** denotes the **presence** and **absence** of $q_i$ in the edited image.

### 3.2 REASONING AND PLANNING: TRANSLATING FAILURES INTO A CORRECTIVE PROMPT

**Quantifying uncertainty of under/over-edits:** We define a two-way uncertainty to capture both edit and preservation fidelity. For each edit requirement failure $q_i \in \hat{Q}$, the agent assigns an uncertainty score $U(q_i) = 1 - sim(b_i, q_i)$, where $b_i$ corresponds to the OWL-ViT generated bounding box in edited image $I_{\text{edit}}$ against $q_i$. A higher $U(q_i)$ indicates stronger evidence that the intended change is absent in $I_{\text{edit}}$. Analogously, for each preservation failure triplet $p_i \in \hat{P}$, we compute: $U(p_i) = 1 - sim(b_i, p_i)$ quantifying the *uncertainty* of $p_i$'s presence in $I_{\text{src}}$. Note, in this case, $b_i$ corresponds to the OWL-ViT generated bounding box in *source image* against $p_i$.

**Prompt refinement:** Having identified a set of preservation failures $\hat{P}$ and a set of edit requirements failures $\hat{Q}$, the *planning module (an LLM)* of `ARTIE` devise a plan to fix them. It takes the following inputs: original edit description $D_{\text{tgt}}$, the set $\hat{P}$, and the set $\hat{Q}$ with their associated uncertainty scores. Through a few-shot prompt, the LLM is instructed to generate a JSON object comprising:

1. **Positive prompt:** The original edit description $D_{\text{tgt}}$ is modified so as it highlights each missing edit $q_i \in \hat{Q}$, and each preservation failure $p_i \in \hat{P}$.

2. **Negative prompt:** To further boost the quality of image generation via the Stable Diffusion model, we also supply negation of each missing edit $q_i \in \hat{Q}$ as well as negation of each preservation failure $p_i \in \hat{P}$, along with their uncertainty scores. These act as a negative prompt.

For more in-depth details about this module, please refer to Section A in the Appendix.

### 3.3 ACTION: THE GUIDED CORRECTION LOOP

Armed with the refined positive prompt and the uncertainty-weighted negative prompt, the agent guides the diffusion model for another attempt, starting from the original (source) image $I_{\text{src}}$. This highly targeted guidance directs the model to correct the errors identified in the previous pass. This entire process—from `SceneDiff` verification (Perception), through uncertainty estimation (Reasoning), prompt refinement (Planning), to the improved generation (Action)—is a closed loop. The newly generated image is fed back to `SceneDiff`, and the cycle repeats until a maximum number of iterations is reached. At the end of each loop, the edited image is evaluated for its fidelity using our reference-free HM-CLIPScore metric (see Section 4.1). The image with the best score across all iterations is selected as the final output ($I_{\text{final}}$). For an example-based illustration of `ARTIE`'s agentic-loop, refer to section A in the Appendix.

Table 1: Results on the Car125, MagicBrush, and GEdit datasets, comparing base image-editing models with `ARTIE` as a plug-and-play module. All metrics except LPIPS are scaled to 0–100. `ARTIE` improvements over base models are shown in blue. Higher values indicate better performance for all metrics except LPIPS.

| Dataset | Model | AugCLIP (↑) | CLIP-T (↑) | CLIP-I (↑) | HM-CLIPScore (↑) | dirCLIP (↑) | LPIPS (↓) |
|---|---|---|---|---|---|---|---|
| *Car125* | *GenArtist* | 69.35 | 25.69 | 84.23 | 39.17 | 2.18 | 0.26 |
| | SDEdit-1.5 | 63.69 | 28.30 | 74.78 | 40.74 | 9.06 | 0.23 |
| | ARTIE + SDEdit-1.5 | 70.97+7.28 | 30.76+2.46 | 88.14+13.36 | 44.79+4.05 | 10.08+1.02 | 0.15-0.08 |
| | SDEdit-3.5 | 75.77 | 29.35 | 92.70 | 44.43 | 10.11 | 0.30 |
| | ARTIE + SDEdit-3.5 | 77.07+1.30 | 30.45+1.10 | 95.06+2.36 | 45.72+1.29 | 11.25+1.14 | 0.24-0.06 |
| | IP2P | 78.01 | 28.03 | 94.35 | 43.07 | 5.10 | 0.16 |
| | ARTIE + IP2P | 79.83+1.82 | 28.57+0.54 | 97.54+3.19 | 43.84+0.77 | 7.67+2.57 | 0.10-0.06 |
| | InstructDiff | 78.85 | 28.37 | 94.62 | 43.48 | 3.19 | 0.20 |
| | ARTIE + InstructDiff | 80.15+1.30 | 29.02+0.65 | 95.64+1.02 | 43.81+0.33 | 5.29+2.10 | 0.18-0.02 |
| | FLUX-Kontext | 78.79 | 28.72 | 95.37 | 43.99 | 9.18 | 0.28 |
| | ARTIE + FLUX-Kontext | 80.23+1.44 | 29.91+1.19 | 97.14+1.77 | 45.24+1.25 | 13.20+4.02 | 0.21-0.07 |
| *MagicBrush* | SDEdit-1.5 | 63.88 | 28.09 | 65.44 | 38.89 | 16.22 | 0.30 |
| | ARTIE + SDEdit-1.5 | 66.57+2.69 | 28.57+0.48 | 77.76+12.32 | 39.90+1.01 | 14.25-1.97 | 0.18-0.12 |
| | SDEdit-3.5 | 73.47 | 26.81 | 81.34 | 40.08 | 11.08 | 0.27 |
| | ARTIE + SDEdit-3.5 | 74.45+0.98 | 27.47+0.66 | 88.91+7.57 | 40.96+0.88 | 11.10+0.02 | 0.18-0.09 |
| | IP2P | 81.05 | 25.92 | 89.43 | 39.87 | 9.14 | 0.17 |
| | ARTIE + IP2P | 85.41+4.36 | 26.62+0.70 | 94.62+5.19 | 40.64+0.77 | 11.45+2.31 | 0.11-0.06 |
| | InstructDiff | 80.96 | 25.51 | 90.10 | 39.51 | 7.07 | 0.16 |
| | ARTIE + InstructDiff | 84.90+3.94 | 26.25+0.74 | 94.42+4.32 | 40.39+0.88 | 9.29+2.22 | 0.12-0.04 |
| | FLUX-Kontext | 84.01 | 25.91 | 91.64 | 40.10 | 9.10 | 0.12 |
| | ARTIE + FLUX-Kontext | 86.00+1.99 | 27.40+1.49 | 94.32+2.68 | 41.38+1.28 | 13.12+4.02 | 0.09-0.03 |
| *GEdit* | IP2P | 78.71 | 25.13 | 88.10 | 38.65 | 6.15 | 0.21 |
| | ARTIE + IP2P | 83.16+4.45 | 26.06+0.93 | 94.54+6.44 | 39.67+1.02 | 9.37+3.22 | 0.15-0.06 |
| | InstructDiff | 78.29 | 24.24 | 87.87 | 37.64 | 4.06 | 0.26 |
| | ARTIE + InstructDiff | 82.19+3.90 | 25.38+1.14 | 93.33+5.46 | 39.03+1.39 | 7.05+2.99 | 0.20-0.06 |
| | FLUX-Kontext | 78.28 | 24.89 | 86.90 | 38.36 | 6.10 | 0.41 |
| | ARTIE + FLUX-Kontext | 81.27+2.99 | 26.19+1.30 | 90.64+3.74 | 39.72+1.36 | 9.12+3.02 | 0.37-0.04 |

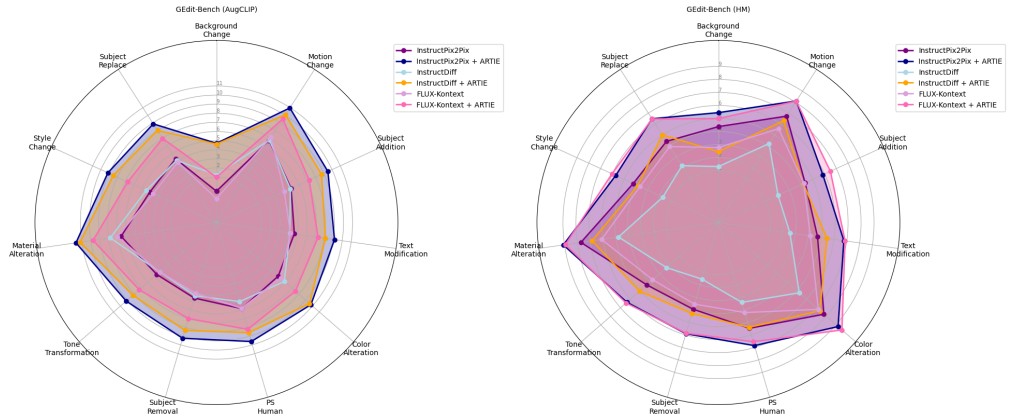

Figure 4: Performance comparison of ARTIE-enhanced versus baseline models on the GEdit dataset across multiple edit categories, demonstrating ARTIE's consistent superiority in fine-grained attribute correction for text-guided image editing

# 4 RESULTS AND ANALYSIS

In this section, we evaluate the performance of our proposed agentic framework, `ARTIE`. We first describe the experimental setup, including datasets, evaluation metrics, and baselines. We then present a quantitative analysis of our results, followed by a qualitative comparison and ablation studies to validate our design choices.

## 4.1 EXPERIMENTAL SETUP

**Datasets:** We evaluate our framework on two distinct datasets to test its performance on both general and complex, multi-aspect editing tasks.

**i) MagicBrush** (Zhang et al., 2024) is a widely recognized benchmark for text-guided image editing. It features a diverse range of images and instructions. However, the edit descriptions are often single-aspect, focusing on one particular change (e.g., "change the color of the car to red"). We use it to validate our method's generalizability.

**ii) Car125** is a challenging dataset we curated to test complex, multi-aspect edits requiring high fidelity. Derived from the publicly available DamageCarDataset containing real post-accident images and damage descriptions, our curation involved: (1) using a VLM (Llama 4.1 Maverick) to generate pre-accident descriptions from post-accident images and damage descriptions (instruction in Section F.5), and (2) synthesizing high-quality pre-accident images using the FLUX text-to-image model from these descriptions. The resulting dataset comprises triplets: (synthetic pre-accident image, pre-accident description, multi-aspect accident description). The task requires applying the complex accident description to the pre-accident image to generate a faithful post-accident image, stress-testing the model's ability to handle multiple fine-grained instructions simultaneously while preserving background elements—a key failure point for existing methods.

**iii) GEdit** (Liu et al., 2025) is a recently introduced benchmark grounded in real-world user instructions. It comprises 1,212 carefully curated instances that reflect authentic editing needs, encompassing diverse scenarios from simple attribute modifications to complex compositional changes (e.g., object additions, scene reconstructions). Unlike synthetic benchmarks, GEdit's instructions vary in complexity and specificity, mirroring genuine user requirements. We use it to evaluate our method's robustness to varied instruction styles and its capability to handle realistic, user-centric editing tasks across multiple difficulty levels.

**Baselines:** As ARTIE is a plug-and-play method, we evaluate it on top of several widely used editing backbones, including SDEdit (Meng et al., 2022) applied to Stable Diffusion-1.5 and Stable Diffusion-3.5, as well as instruction-following editing models such as InstructPix2Pix (aka IP2P) (Brooks et al., 2023), InstructDiffusion (Geng et al., 2023), and FLUX-Kontext (Labs et al., 2025).

In addition, we include *GenArtist* (Wang et al., 2024a) as an agentic baseline on the CAR125 dataset and compare ARTIE directly against it.

**Evaluation metrics:** We evaluate editing quality using standard metrics: *AugCLIP*, *CLIP-T*, and *CLIP-I* (Hessel et al., 2022). CLIP-T measures alignment with the target instruction, and CLIP-I measures source-image preservation; AugCLIP provides a balanced fidelity score. *HM-CLIPScore*, the harmonic mean of CLIP-T and CLIP-I, summarizes edit faithfulness and preservation and also serves as the stopping criterion of our correction loop. We additionally report *LPIPS*, which measures perceptual similarity between the source and edited image, lower values indicate better preservation. *dirCLIP* (Gal et al., 2021) captures whether the semantic direction of the edit is correct. All metrics range from 0 to 1, with higher values indicating better edit quality, except LPIPS, which ranges from 0 to $\infty$ and where lower values indicate better preservation.

## 4.2 QUANTITATIVE ANALYSIS

As shown in Table 1, the proposed agentic framework, ARTIE, demonstrates consistent performance enhancements when applied as a plug-and-play module across diverse base architectures, revealing several key insights regarding its design and effectiveness.

**Model-agnostic enhancement capability.** The framework exhibits performance improvements regardless of underlying architecture, encompassing diffusion-based models (SDEdit-1.5, SDEdit-3.5), instruction-tuned systems (IP2P, InstructDiff), and rectified flow models (FLUX-Kontext). This architecture-agnostic behavior suggests that ARTIE's agentic verification mechanisms address fundamental image editing limitations that transcend specific model designs, rather than merely compensating for architecture-specific weaknesses.

**Balanced improvement across complementary metrics.** The framework simultaneously enhances both text-alignment metrics (CLIP-T, AugCLIP) and structural preservation metrics (CLIP-I, LPIPS),

Table 2: Ablation study of our agentic components on the Car125 dataset, applied to both SDEdit-3.5 and SDEdit-1.5. IPR refers to the LLM-based initial prompt refinement, and 'cutoff' indicates the stopping condition for the multi-cycle loop. A higher score indicates better performance.

| Model Configuration | AugCLIP (↑) | CLIP-T (↑) | CLIP-I (↑) | HM-CLIPScore (↑) | dirCLIP (↑) |
|---|---|---|---|---|---|
| *Stable Diffusion 3.5 Ablations* | | | | | |
| SDEdit-3.5 (base) | 75.77 | 29.35 | 92.70 | 44.43 | 0.10 |
| + IPR Only | 76.98 | 29.75 | 92.94 | 44.89 | 0.10 |
| + Agentic (cutoff@MAX), no IPR | 76.81 | 29.47 | 93.66 | 44.70 | 0.10 |
| + Agentic (cutoff@MAX), with IPR | 77.03 | 29.65 | 93.67 | 44.92 | 0.10 |
| + Agentic (cutoff@BEST), no IPR | 76.30 | 30.40 | **95.19** | 45.70 | 0.10 |
| **+ Agentic (cutoff@BEST), with IPR [Ours]** | **77.07** | **30.45** | 95.06 | **45.72** | **0.11** |
| *Stable Diffusion 1.5 Ablations* | | | | | |
| SDEdit-1.5 (base) | 63.69 | 28.30 | 74.78 | 40.74 | **0.09** |
| + IPR Only | 70.04 | 29.75 | 83.55 | 43.65 | 0.08 |
| + Agentic (cutoff@MAX), no IPR | 65.54 | 27.89 | 77.94 | 40.81 | 0.07 |
| + Agentic (cutoff@MAX), with IPR | 70.50 | 29.02 | 83.69 | 42.88 | 0.06 |
| + Agentic (cutoff@BEST), no IPR | 67.25 | 30.00 | 84.18 | 43.22 | 0.07 |
| **+ Agentic (cutoff@BEST), with IPR [Ours]** | **70.97** | **30.76** | **88.14** | **44.79** | 0.07 |

thereby circumventing the conventional trade-off wherein improvements in one dimension occur at the expense of another. The substantial LPIPS reductions observed alongside meaningful semantic alignment gains indicate that agentic verification successfully navigates the inherent tension between faithful edit execution and background preservation.

**Robustness across dataset characteristics.** Consistent improvements observed across domain-specific (Car125), general-purpose (MagicBrush), and diverse editing scenarios (GEdit) demonstrate strong generalization capabilities. Notably, ARTIE yields substantial gains even when base models already exhibit strong standalone performance, indicating that scene-based decomposition and iterative verification capture fundamental principles of structured image editing that remain valuable across varying dataset complexities.

These findings validate that training-free agentic orchestration can systematically enhance image editing systems without requiring domain-specific fine-tuning or architectural modifications.

**Fine-grained editing analysis.** Fig. 4 shows that ARTIE consistently improves baseline models across all edit categories in GEdit, including style, material, subject, motion, background, and text edits. ARTIE-enhanced models achieve higher attribute-level accuracy and fidelity throughout, demonstrating its effectiveness for fine-grained image editing on a standardized benchmark.

**Ablation studies:** We conduct ablation studies on the Car125 dataset to validate the contributions of our framework's key components, with results presented in Table 2.

*Impact of Prompt Refinement (IPR):* Initial LLM-based prompt refinement (+IPR Only) provides a consistent, albeit modest, improvement over the base models. For SDEdit-1.5, IPR alone boosts the HM-CLIPScore from 40.74 to 43.65, showing its utility as a strong starting point for the agentic loop.

*Impact of Agentic Loop:* The core agentic feedback mechanism provides significant performance gains. When applied without IPR (+ Agentic, no IPR), the framework substantially improves scores, particularly the crucial CLIP-I preservation metric. This highlights the effectiveness of the SceneDiff verification pipeline in correcting editing errors.

*Stopping Criterion:* The best performance is achieved when all components work in synergy. Furthermore, the choice of stopping criterion is critical. Using our reference-free HM-CLIPScore to select the best edit from the correction loops (cutoff@BEST) consistently outperforms running for a fixed maximum number of loops (cutoff@MAX). For our proposed ARTIE +SDEdit-1.5, this strategy elevates the HM-CLIPScore from 42.88 to 44.79, and increases the proposed ARTIE +SDEdit-3.5 score from 44.92 to 45.72. This illustrates that selecting the optimal edit cycle via a reference-free metric is essential for balancing fidelity and preservation.

## 4.3 QUALITATIVE ANALYSIS

Figure 5 showcases ARTIE's superior performance on complex editing tasks from our Car125 dataset, comparing it against baselines (i.e. InstructPix2Pix, InstructDiffusion, and SDEdit-3.5) in two scenarios. We compare ARTIE+ SDEdit-3.5 with the aforementioned baselines.

**Edit Description:** The image shows a close-up view of a car part, likely a section of the car's body or bumper. The surface is light-colored, possibly white or light gray, and has visible damage. The **damage is labeled as a "dent,"** which is evident from the **indentation in the metal.** The dent is surrounded by **some scratches and dirt**, indicating that the area has been impacted. The surrounding area has some **minor scuffs and marks**, suggesting the car has been in a minor collision or has been struck by an object.

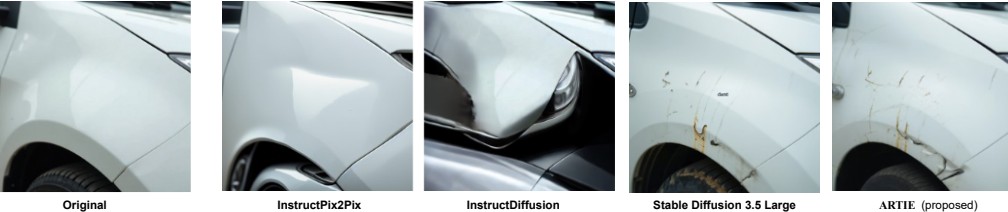

**Edit Description:** The image shows a close-up of the rear section of a car, specifically focusing on the bumper and the area around the rear wheel. The **damage is labeled as 'scratch,'** and the scratches are prominent, with some areas showing **exposed metal or underlying layers of paint.** The bumper also has a **noticeable dent and paint chipping**, indicating a significant impact.

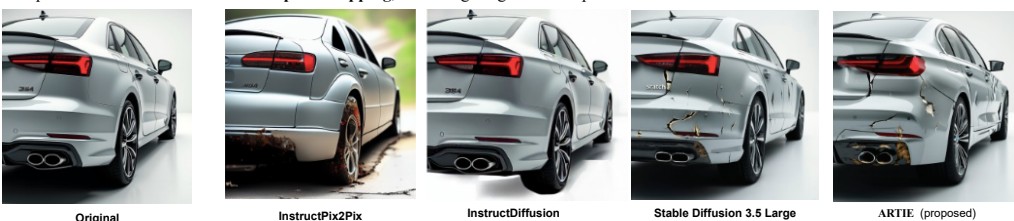

Figure 5: Qualitative comparison of our method against baselines on the Car125 dataset. Baselines were provided with shortened prompts to accommodate their input length constraints. Edit aspects are given in **bold** font.

In the top row, the task is to add a 'dent with scratches and dirt.' InstructPix2Pix renders only a subtle dent, while Stable Diffusion 3.5 adds minor scuffs but misses the dent. InstructDiffusion suffers from extensive over-editing, distorting the car. In contrast, ARTIE successfully renders a prominent dent with surrounding textures, faithfully executing the prompt.

The bottom row features a more complex instruction: 'a noticeable dent, paint chipping, and scratches with exposed metal.' Here, InstructPix2Pix and InstructDiffusion fail completely, hallucinating a new car and background. Stable Diffusion 3.5 applies some scratches but misses the other effects. ARTIE is the only method to correctly apply all three damages while maintaining near-perfect fidelity to the original car and scene. For more examples, see Figure 10 in the appendix.

**Edit Description:** replace the fork with a knife

**Edit Description:** make a lion laying on the couch

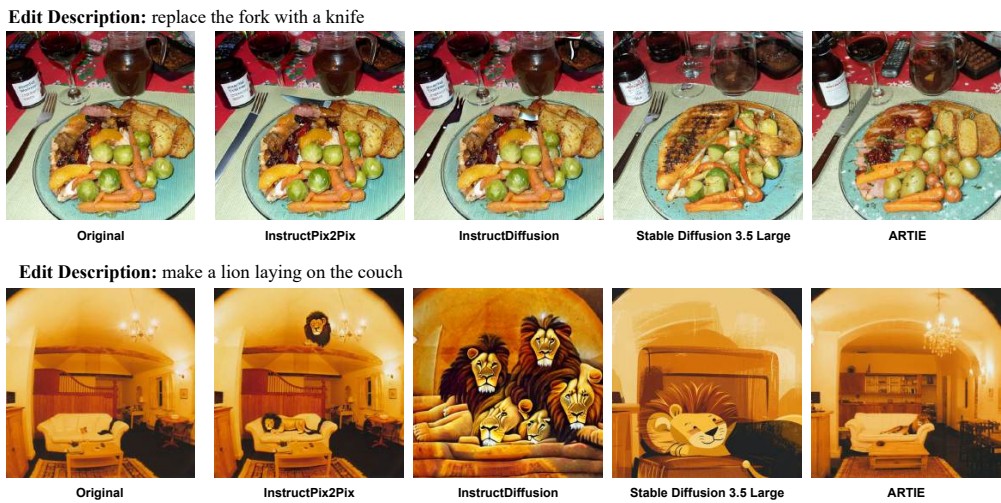

Figure 6: Examples of images edited using baseline models and ARTIE on MagicBrush.

**Edit Description:** The image shows a close-up of a car's front bumper, which is painted in a light blue color. There is visible damage labelled as a "scratch." The scratch is **extensive, running along the edge of the bumper**, with the paint chipped off in several areas, exposing the underlying material. The **damage appears to be significant**, affecting the aesthetic appearance of the bumper. The car's tire and part of a headlight or fog light are also visible in the image.The **surface around the scratch is relatively smooth**, indicating that the damage is primarily superficial.

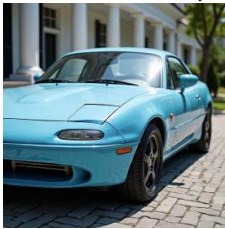 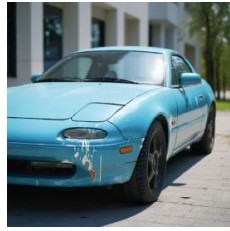 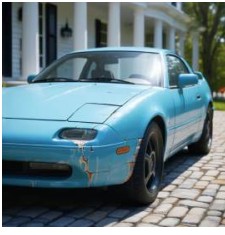 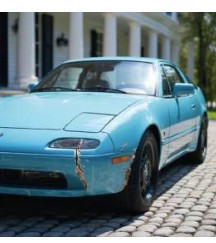

| Original | Stable Diffusion 3.5 Large | Stable Diffusion 3.5 Large + IPR | ARTIE (Proposed) |

Figure 7: Qualitative example concerning `ARTIE` vs Initial Prompt Rewriting (IPR) on SDEdit-3.5.

Figure 6 further demonstrates `ARTIE`'s generalizability on the MagicBrush dataset, where it surpasses baseline models that are explicitly fine-tuned on the MagicBrush dataset. For example, given the instruction to 'replace the fork with a knife,' the naive SDEdit-3.5, along with other baseline models, fail to fully transform the fork, leaving portions unaltered as shown in the comparative images. `ARTIE` +SDEdit-3.5, however, successfully performs the complete substitution, yielding a coherent knife in place of the fork. Similarly, when tasked to 'make a lion laying in the couch,' all baseline models create multiple or cartoon-like lions. In contrast, `ARTIE` adheres to the compositional constraint of the edit description by properly generating a single, realistic lion as instructed. For more visual examples, refer to Figure 11 in the appendix.

While these examples highlight cases where `ARTIE` outperforms existing baselines, its agentic design, built on multiple verifier modules, also introduces potential failure modes. Errors originating in any module may propagate through the editing loop, occasionally producing unintended edits in the final output. We identify two such categories of errors and discuss them in detail in Appendix C. Furthermore, the iterative agentic loop in `ARTIE` introduces additional overhead beyond standard editing pipelines. Appendix D presents detailed runtime and resource measurements.

**Can `ARTIE` improve over initial prompt rewriting?** `ARTIE` is essentially a mechanism that improves the prompt to increase the fidelity of the edited image eventually. In image 7, we show such examples where it can be seen that, on qualitative aspects of the prompt (highlighted in bold), the `ARTIE` generated picture is better than Initial Prompt Rewrite (IPR). The base model's edit fails to render key details from the prompt. IPR improves this, creating a more extensive scratch that begins to expose the underlying material. However, `ARTIE`'s iterative feedback loop achieves superior fidelity. The final output of `ARTIE` renders a more prominent scratch while maintaining a smoother surrounding surface, keeping the scratch aligned along the edge of the bumper, and more accurately capturing the prompt's intent. This qualitative improvement is confirmed by the HM-CLIPScore, which rises from 39 (SDEdit-3.5 as a baseline) to 40 (IPR), and finally to 41 for the `ARTIE`.

## 5 Limitations, Conclusions, and Future Work

We introduced `ARTIE`, a plug-and-play, inference-time agentic framework that couples a perception module, `SceneDiff`, with an LLM-driven prompt engineer to iteratively refine diffusion-based edits without retraining. Evaluations on Car125 and MagicBrush show consistent gains in multi-aspect fidelity and background preservation.

`ARTIE` depends on external perception modules (*e.g.*, OWL-ViT and CLIP) and LLM/VLM, so errors or biases in these tools can propagate through the loop. The multi-cycle verification and regeneration increase the latency and memory requirement. The uncertainty-weighted negative prompts are heuristic-based and, therefore, they may cause mis-handling of subtle attributes.

In our future endeavors, we plan to integrate stronger grounded verifiers (*e.g.*, segmentation-level checks) and learned uncertainty calibration to reduce false corrections. Further, latency can be improved via caching, early stopping, batched verification, and utilizing lightweight backbones.

ETHICS STATEMENT

This work is conducted entirely on publicly available datasets, and all models employed are open-sourced through the Hugging Face platform. Consequently, the research is designed to be transparent, reproducible, and consistent with the usage policies governing the underlying resources. No proprietary or personally identifiable data is introduced at any stage of experimentation.

Nevertheless, text-guided image editing technologies raise non-trivial ethical concerns. The most critical among these is the potential for misuse in the generation of manipulated or deceptive media, including so-called deepfakes. The capacity to synthesize realistic yet fabricated content poses risks of disinformation, reputational harm, and erosion of trust in digital artifacts. While the intent of this work is to advance methods for controllable and creative visual editing, the dual-use nature of the technology is acknowledged. Importantly, a substantial body of concurrent research is devoted to the detection and mitigation of synthetic media, including deepfake detection and watermarking techniques, which provides a counterbalance to the risks associated with misuse.

In addition, reliance on existing datasets and foundational models introduces the possibility of amplifying social biases encoded in the training distributions. Text-driven editing may interact with such imbalances in ways that reinforce stereotypes or produce systematically skewed outputs. Although our approach does not contribute new datasets, practitioners deploying these methods should remain attentive to dataset composition and model behavior in order to avoid unintended harms.

A further consideration pertains to environmental impact. While our method leverages pretrained models and thereby avoids the computational and energy costs of training from scratch, inference at scale still carries non-negligible resource requirements. Given the increasing ubiquity of generative models, the cumulative carbon footprint of their deployment warrants serious attention.

In summary, while our work demonstrates technical progress in controllable image editing, responsible use requires continued attention to issues of misuse, bias, and sustainability.

REPRODUCIBILITY STATEMENT

To ensure reproducibility, we adopt a fixed random seed of $42$ across all experiments. All datasets used in this work are publicly available, and all models are obtained from Hugging Face. The experimental setup and methodological details are described in detail in sections 3, A and 4, providing all necessary information for replication. Upon acceptance, we will release the complete source code together with scripts for evaluation, as well as representative model outputs corresponding to the reported results. These outputs can be cross-verified with results generated from the released codebase, thereby ensuring reproducibility.

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

# A AN END-TO-END EDITING LOOP OF ARTIE VIA EXAMPLE

## A.1 FORMATION OF PRESERVATION FAILURE TRIPLETS

The process shown in the figure 8 illustrates how ARTIE's scene graph-based preservation analysis operates for the prompt 'make a lion lay on couch.' We begin with an initial edited image synthesized using a generative model (such as Stable Diffusion), which might contain editing errors.

(1) Starting from the original (source) image, a scene graph is generated by the Vision Language Model (LLaMA-4 Maverick) using the edit description, taken as input, and provided to the VLM using a specific source scene graph prompt (refer to Section F.1). This source scene graph encodes objects, attributes, and relationships present in the original image, for example, elements like 'room has couch,' 'rug is colorful,' or 'walls are white.'

(2) Next, we construct a target scene graph prompt (refer to F.2) by utilizing the source scene graph obtained in step 1. This target scene graph prompt, along with an intermediate image, is provided to the VLM that generates the target scene graph. The target scene graph captures the atomic facts that should be true in the edited image. In this example, it will contain visual elements such as 'room has lion,' 'lion is on bed', 'bed is large', or 'lion is sleeping'.

The source scene graph and target scene graph are output as plain texts, which are a list of triplets in the form of subject-verb-object. This triplet structure enables their object properties to be compared in a one-to-one manner.

(3) Finally, a comparison is carried out between the source and target scene graphs, typically by set difference. This identifies the 'preservation failure triplets': triplets present in the original but either missing or incorrectly modified in the edited image. The list of these triplets precisely details which parts of the original scene were not preserved, thus providing explicit, structured feedback on over-editing and content loss.

In summary, the numbered arrows correspond to a stepwise process in which both the original and the initially edited images, as well as the textual edit description, are parsed into structured representations by a VLM, then compared to extract detailed, explicit feedback about what content has been lost or inappropriately changed due to the edit.

## A.2 PERCEPTION–REASONING–ACTION LOOP

Figure 9 illustrates the second phase of SceneDiff, which realizes the perception–reasoning–action loop for iterative correction.

**Perception (Verification):** The process begins with aspect extraction from the edit instruction (e.g., 'make a lion lay on couch') using an aspect extraction prompt (refer to F.3), where the Vision–Language Model (LLaMA-4 Maverick) generates atomic requirements such as 'there is a lion' and 'the lion is laying on the couch.' For each aspect, a rule-based augmentation creates a corresponding negative variant (e.g., 'there is not a lion'; 'the lion is not laying on the couch'). These aspects are then paired with the intermediate image and analyzed by OWL-ViT, which localizes candidate regions potentially reflecting the specified aspects. Each region, along with the positive and negative variants, is evaluated using CLIP. CLIP computes embedding similarities to produce confidence scores for both variants. If the negative variant achieves a higher score than the positive one, the aspect is considered missing in the current edit and is added to the unedited-aspect list. These are called 'edit-requirement failures'.

**Reasoning and Planning:**

1. First, we compute uncertainty scores to evaluate both the fidelity of edits and the preservation of content. For each failed edit requirement, we calculate an uncertainty score to assess how accurately the intended change appears in the edited image. A higher score suggests that the intended change is likely absent. Similarly, for each preservation failure, we calculate an uncertainty score to gauge whether the source image contains the content. A higher score indicates greater uncertainty about whether the source image includes the desired content. In Figure 9, it is done using **uncertainty estimation module**.

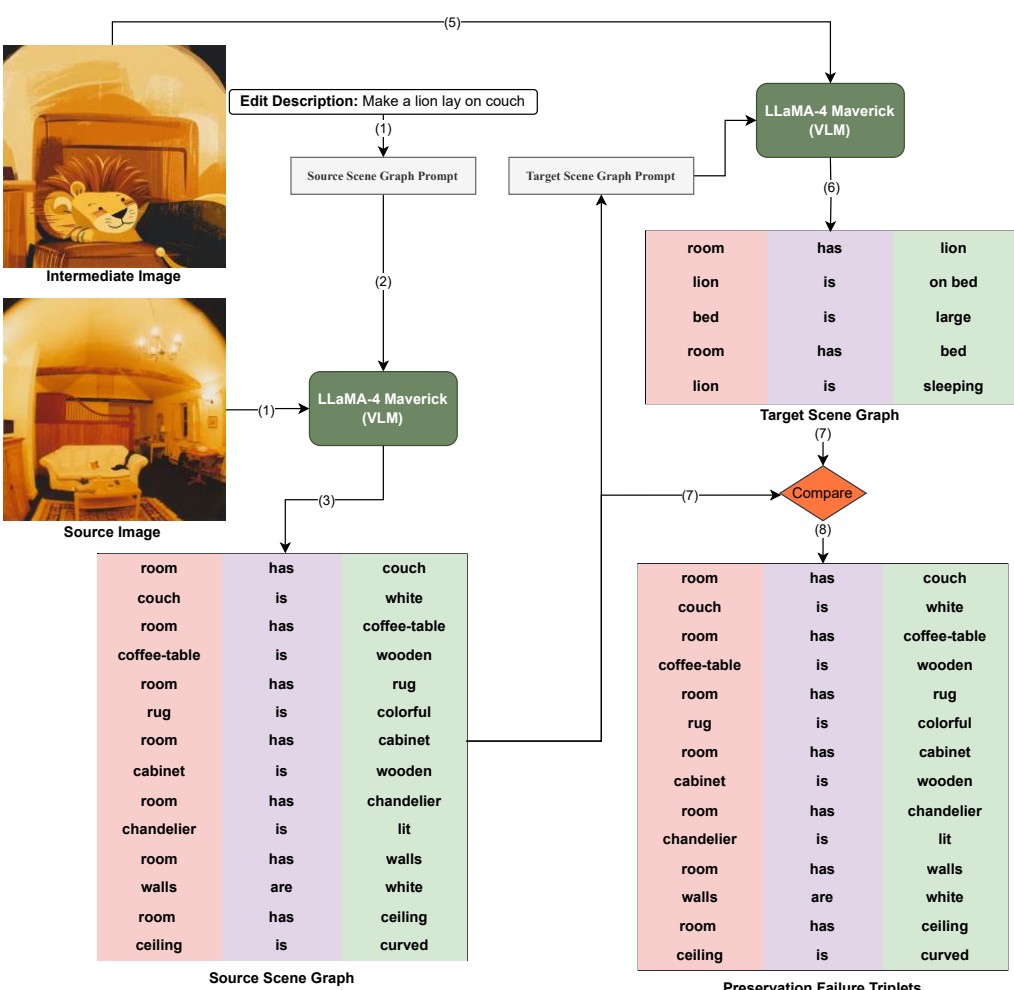

**SceneDiff Phase 1:** Detecting over-editing with Scene Graph Alignment.

Figure 8: Workflow for formation of preservation failure triplets in ARTIE's agentic editing loop. Given an edit prompt and a source image, structured scene graphs are extracted for both the original and edited images using a vision-language model. Comparison of these graphs exposes "preservation failure triplets," explicitly indicating objects, attributes, or relationships that should remain unchanged but have been lost or improperly edited. This enables fine-grained feedback about over-editing and content loss to drive subsequent correction cycles.

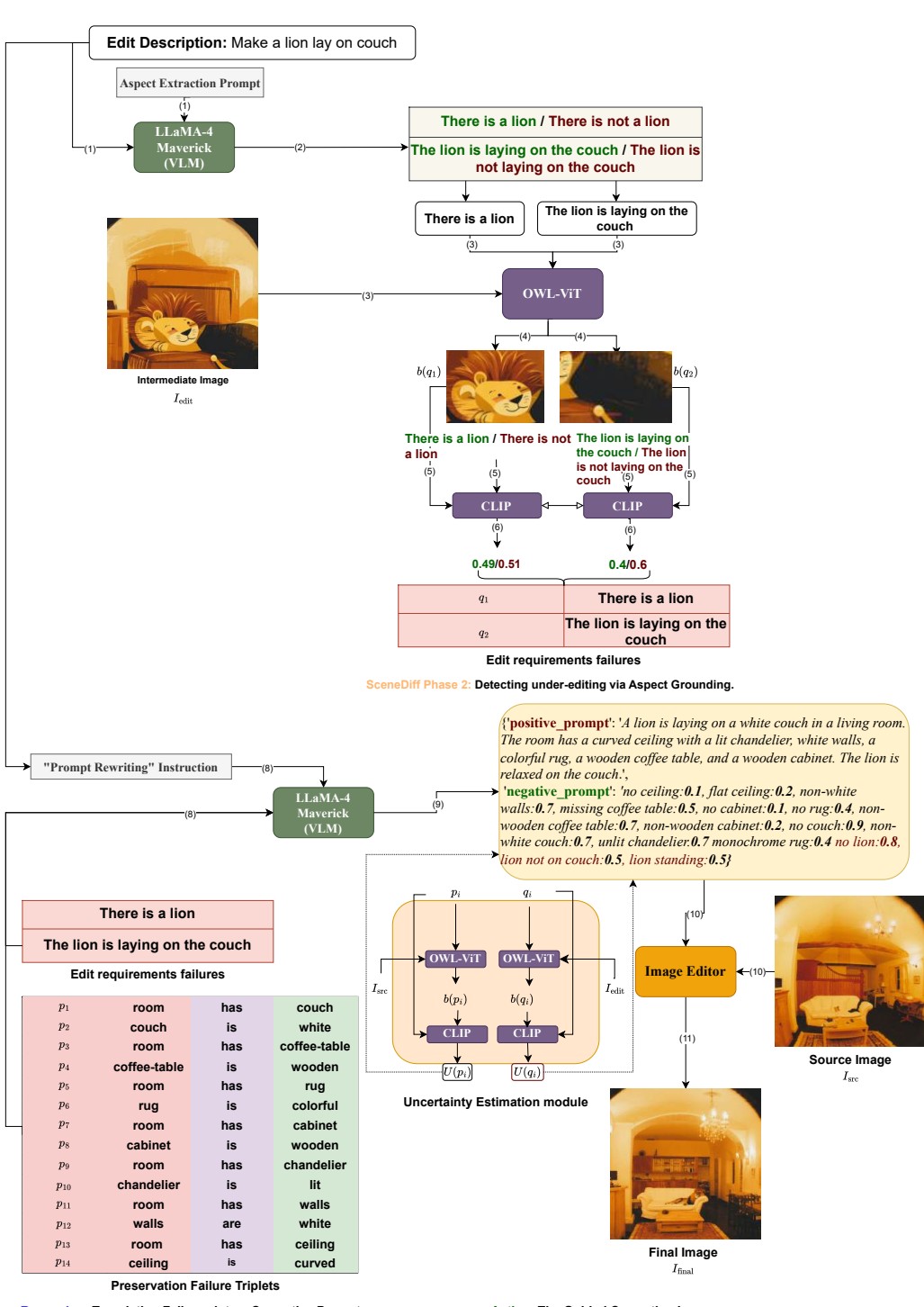

Figure 9: Illustration of the perception–reasoning–action loop in ARTIE. Atomic requirements are extracted from the edit instruction and verified for presence in the intermediate image using OWL-ViT and CLIP, quantifying missing or incorrect aspects. These failures, together with preservation failure triplets from scene graph analysis, are passed to an LLM-based prompt engineer, which synthesizes a refined positive prompt and an uncertainty-weighted negative prompt. The prompts guide the next image editing cycle, enabling iterative correction until the output matches the intended scene description while preserving contextual content.

2. The set of failed edit requirements, together with the original edit instruction and preservation failure triplets, is passed to the reasoning module. Here, an LLM-based prompt engineer (LLaMA-4-Maverick) reformulates them, using a prompt-rewriting instruction (refer to Section F.4) into a structured corrective positive prompt. The output includes a refined positive prompt that explicitly specifies the intended scene (*e.g.*, 'A lion is laying on a white couch in a living room. The room has a curved ceiling with a lit chandelier, white walls, a colorful rug, a wooden coffee table, and a wooden cabinet. The lion is relaxed on the couch.'), as well as an uncertainty weighted negative prompt that penalizes missing or incorrect elements (*e.g.*, 'no ceiling,' 'flat ceiling,' 'non-white walls,' 'missing coffee table,' 'no lion,' 'lion not on couch').

**Action (Correction):** The refined positive and negative prompts are then provided to the image editor along with the source image for another generation cycle, producing a new edited image that incorporates the corrections. This loop continues iteratively, with each round informed by verification scores and preservation failure triplet feedback, until the system converges to a high-fidelity result that closely aligns with the intended textual edit while maintaining the broader context of the original scene.

# B  ADDITIONAL QUALITATIVE EXAMPLES

**Car125:**  Figure 10 showcases a series of qualitative comparisons, demonstrating ARTIE's superior performance on the Car125 dataset against baseline models like InstructPix2Pix, InstructDiffusion, and Stable Diffusion 3.5 Large. Each row presents a complex, multi-aspect car damage editing task where ARTIE excels in both fidelity and preservation.

In the first example (Figure 10a), the instruction is to add heavy dents, scratches, and cracks to the front of a white sports car. While InstructPix2Pix changes the car's identity and InstructDiffusion severely distorts it, ARTIE is the only model to apply realistic, heavy damage (especially on the bumper) while preserving the original vehicle's structure. Moreover, the severity of the accident is reflected better in ARTIE compared to SDEdit-3.5-Large.

Figure 10b requires adding prominent scratches and peeling paint to a rear bumper. The baselines either produce minimal scratches (InstructPix2Pix), unrealistic artefacts (InstructDiffusion), or large, crack-like patterns (Stable Diffusion). In contrast, ARTIE successfully renders fine-grained scratches with patches of peeled paint of green colour, precisely matching the textual description.

In Figure 10c, which involves adding cracks to a headlight and scuffs to the bumper, ARTIE again demonstrates superior fine-grained control. It accurately adds both the cracks on the lens and the blue paint marks on the bumper, whereas the other models fail to apply both edits correctly or introduce unwanted stylistic changes. For example, InstructDiffusion changes the car colour (over-edit), InstructPix2Pix makes minimal change (under-edit), and SDEdit-3.5-Large adds a text *Lamp Broken* below the headlamps (under-edit).

Finally, in Figure 10d, when tasked with adding scratches and dents to the rear bumper, InstructPix2Pix and InstructDiffusion fail to add any damage, merely making the tail light glow. SDEdit-3.5-Large adds unrealistic smudges, but ARTIE correctly applies both scratches and a visible dent, demonstrating its ability to handle multiple simultaneous damage types.

Across all examples, ARTIE consistently adheres to complex instructions with high fidelity, successfully rendering multiple specified damages while maintaining the identity and context of the original scene, outperforming the baseline models that exhibit common failures like under-editing, over-editing, or loss of context.

**MagicBrush:**  Figure 11 illustrates qualitative comparisons of our method, ARTIE, versus multiple baseline models on diverse, complex text-guided image editing tasks. Each row contains the original image, baseline model edits, and the ARTIE-generated result for a specific edit instruction. ARTIE consistently produces edits that better adhere to the textual instructions, avoiding common failures seen in baselines such as under-editing key regions, adding unintended objects, or hallucinating multiple instances of single objects. For example, in Figure 11a, the requirement is replacing the wooden floor with tiles, ARTIE cleanly replaces the flooring texture, while the baseline models (despite being trained on this particular dataset) retain wooden textures or introduce artefacts. In the wine bottle addition task (in Figure 11b), baseline methods duplicate bottles inaccurately, whereas

**Edit Text:** The image shows the front part of a car that has sustained **significant damage**. The headlight on the left side is damaged, and the **surrounding area is heavily dented and scratched.** The **bumper and the grille are also damaged**, with visible cracks and deformations. The **damage is extensive** and goes beyond just a scratch, indicating a more severe impact. The car's paint is chipped and there are visible signs of wear and tear around the affected area.

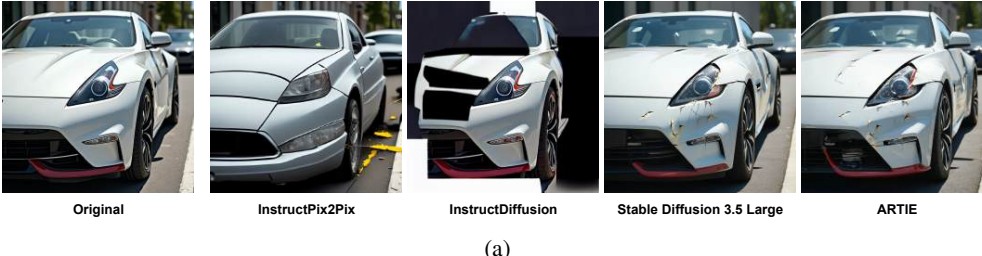

Original     InstructPix2Pix     InstructDiffusion     Stable Diffusion 3.5 Large     ARTIE

(a)

**Edit Text:** The image shows the rear section of a car, specifically the bumper area. The **bumper has visible damage, including scratches and paint peeling off.** The scratches are prominent, with some areas showing **exposed metal or primer beneath the paint.** The damage is concentrated on the right side of the bumper, near the tail light. The car's paint appears to be a dark color, **possibly green or gray.**

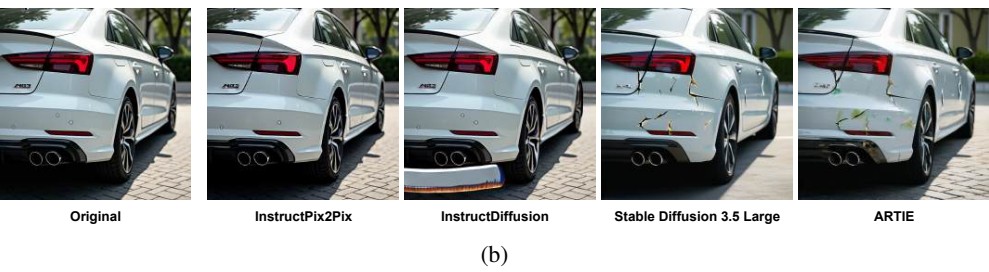

Original     InstructPix2Pix     InstructDiffusion     Stable Diffusion 3.5 Large     ARTIE

(b)

**Edit Text:** The image shows a close-up view of the front part of a car, specifically focusing on the headlight area. **The headlight is visibly damaged, with cracks and scratches on the lens.** There are also **blue paint marks and scuffs on the surrounding area of the car's bumper**, indicating some form of collision or impact. The damage is labeled as **'lamp broken,'** highlighting the broken state of the headlight. The car's body is a light color, possibly silver or gray.

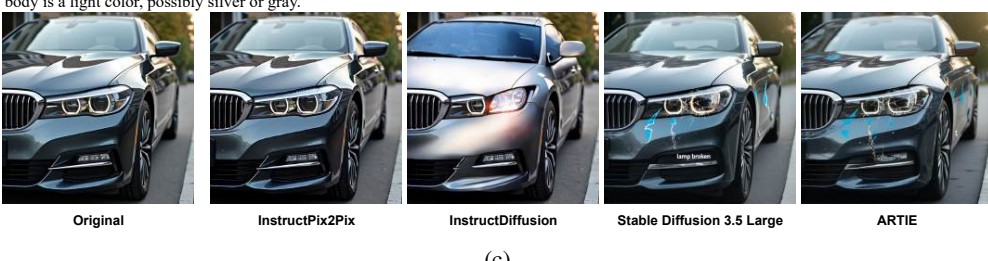

Original     InstructPix2Pix     InstructDiffusion     Stable Diffusion 3.5 Large     ARTIE

(c)

**Edit Text:** The image shows the rear end of a car, specifically focusing on the left tail light and bumper area.**The bumper below the tail light also shows signs of damage, including scratches and dents.** The car is parked in an outdoor area, with buildings visible in the background.

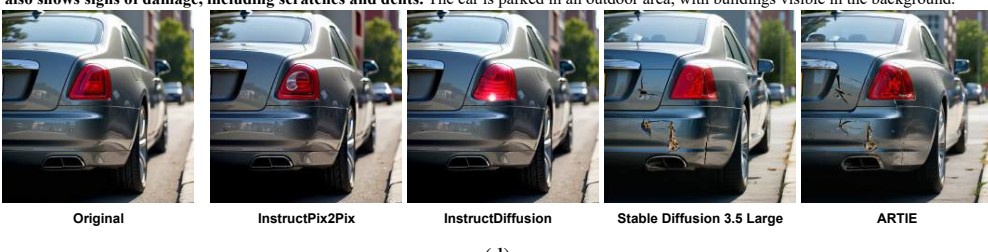

Original     InstructPix2Pix     InstructDiffusion     Stable Diffusion 3.5 Large     ARTIE

(d)

Figure 10: Examples from the Car125 dataset where ARTIE beats standard Stable Diffusion 3.5 Large and surpasses standard baselines. Note that these baselines are not trained for this task. Thus, the Car125 dataset constitutes out-of-domain examples, leading to either over-edit or under-edit in the baseline images.

ARTIE correctly adds a single bottle. Similarly, when tasked to 'replace the chicken with rice on the plate,' all baseline models incorrectly preserve remnants of the original food items. In contrast, ARTIE adheres to the compositional constraint by fully converting the plate's contents to rice, as illustrated in 11c. Moreover, in Figure 11d, ARTIE successfully generates a tower where others

**Edit Description:** change the wooden flooring into tile flooring

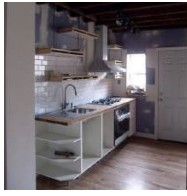 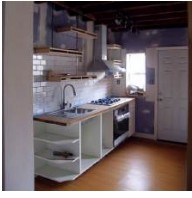 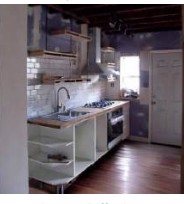 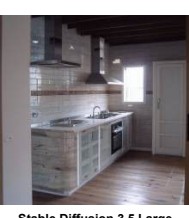 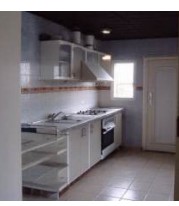

Original     InstructPix2Pix     InstructDiffusion     Stable Diffusion 3.5 Large     ARTIE

(a) The original image shows a kitchen with wooden flooring, and the edit instruction requires the wooden floor to be changed to a tile floor. Only ARTIE follows the instruction faithfully.

**Edit Description:** What if there is a bottle of wine on the table?

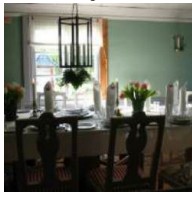 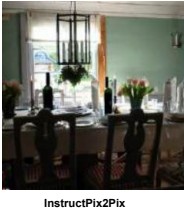 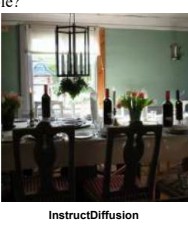 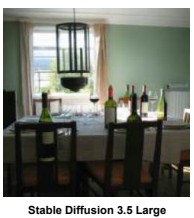 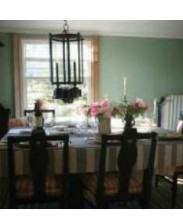

Original     InstructPix2Pix     InstructDiffusion     Stable Diffusion 3.5 Large     ARTIE

(b) Baseline models add multiple wine bottles, while ARTIE only places one wine bottle, adhering to the edit description better.

**Edit Description:** change the chicken, broccoli and cheese into rice

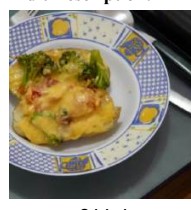 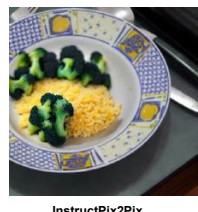 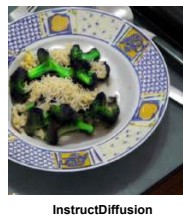 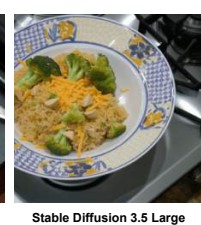 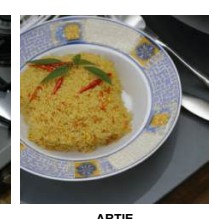

Original     InstructPix2Pix     InstructDiffusion     Stable Diffusion 3.5 Large     ARTIE

(c) InstructPix2Pix and InstructDiffusion retain remnants of food items. ARTIE properly modifies the food to rice.

**Edit Description:** change the brick wall into a tower

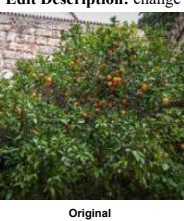 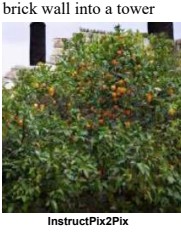 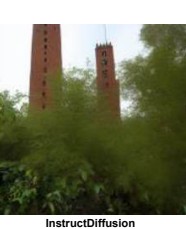 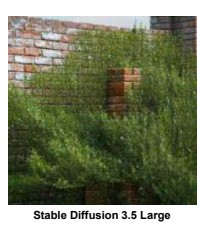 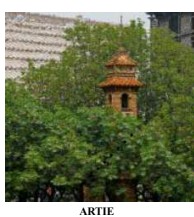

Original     InstructPix2Pix     InstructDiffusion     Stable Diffusion 3.5 Large     ARTIE

(d) InstructPix2Pix preserves the bush in the original image but fails to add a tower. The rest of the methods fail to preserve the original scene.

**Edit Description:** add a soda can in the back

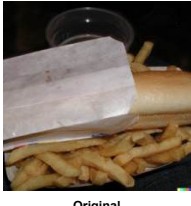 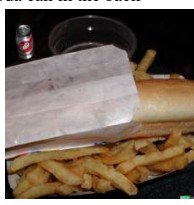 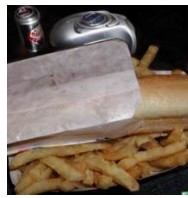 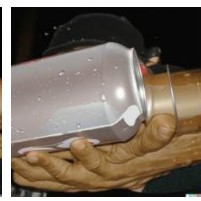 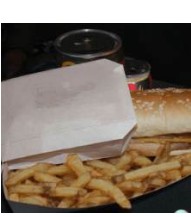

Original     InstructPix2Pix     InstructDiffusion     Stable Diffusion 3.5 Large     ARTIE

(e) ARTIE correctly places a soda can in the back while stable diffusion fails. InstructPix2Pix performs the best fidelity edit.

Figure 11: Examples from MagicBrush: In general, ARTIE exhibits better adherence to edit description while preserving the scene details. Note that these baselines are especially trained for this task.

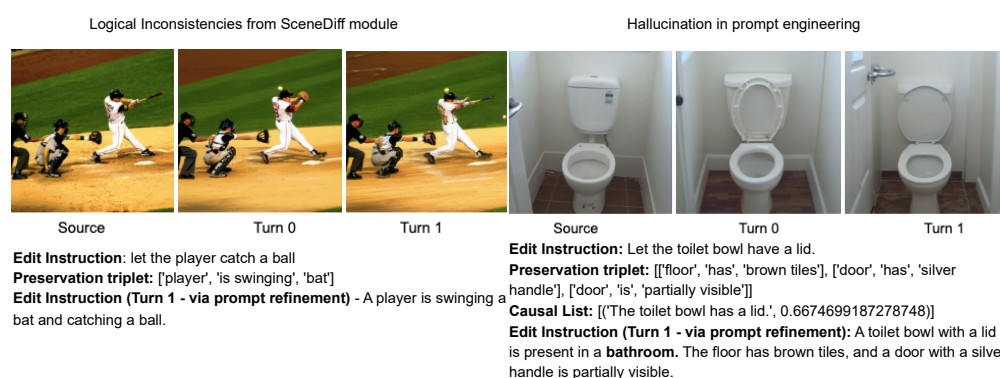

Figure 12: Error Analysis across two categories taking SDEdit 3-.5 as the base model on the MagicBrush dataset.

fail to clearly insert the structure or produce blurry results. Finally, `ARTIE` places a soda can in the requested location with higher fidelity and clarity (refer to Figure 11e) than competing methods, demonstrating precise adherence to complex, multi-aspect instructions.

## C  ERROR ANALYSIS

While ARTIE consistently improves multi-aspect fidelity across datasets, its performance is shaped by the behaviour and limitations of the verifier stack that underpins the SceneDiff module. In this section, we examine two representative failure modes observed during iterative refinement: (i) logical inconsistencies arising from scene-graph construction, and (ii) hallucinations introduced during prompt engineering. Figure 12 illustrates this.

**1. Logical Inconsistencies from SceneDiff.**  SceneDiff constructs preservation triplets by instructing the VLM to extract all scene elements that should remain unchanged, explicitly excluding any concept related to the causal edit. However, in some cases the VLM violates this constraint and mistakenly includes aspects that conflict with the requested modification.

A prominent example occurs for the instruction *"let the player catch a ball"*. The preservation set generated after the first iteration incorrectly contains the triplet *(player, is swinging, bat)*—an action mutually exclusive with ball-catching. When the planning module subsequently incorporates this triplet into the refined prompt, the resulting instruction becomes internally contradictory ("a player is swinging a bat and catching a ball"), causing the diffusion model to synthesize an implausible scene. This demonstrates how VLM-level inconsistencies can propagate into the agentic loop, yielding edits with incompatible semantics.

**2. Hallucination in Prompt Engineering.**  A second failure mode emerges from the LLM-based refinement stage, where contextual inference occasionally exceeds what is grounded in either the causal or preservation lists. For instance, given the instruction *"let the toilet bowl have a lid"*, the preservation triplets correctly include elements such as floor tiles and a partially visible door, and the causal list contains only the addition of a lid. However, the refined prompt produced by the planning module expands this into a higher-level scene description: *"A toilet bowl with a lid is present in a bathroom."*

Although semantically reasonable, this inferred notion of a "bathroom" is not present in the source image and may trigger the generation of additional objects (e.g., a fully visible door and door handles). These hallucinated elements alter the global layout of the scene, producing unintended changes that are neither grounded in the original visual content nor required by the instruction. Initial turn image is more aligned towards the edit instruction and additional context brought by the prompt refinement does more harm than good in the turn 1 image. This illustrates a limitation of using large

language models for prompt refinement: while capable of producing coherent descriptions, they may over-generalize based on prior knowledge, inadvertently activating scene priors in the diffusion model.

**Summary.** These errors highlight inherent limitations of external verifiers within the agentic loop. Logical contradictions arise primarily from VLM misalignment during preservation-triplet extraction, whereas unwanted contextual hallucinations originate from the LLM's tendency to inject inferred scene semantics. Both failure modes underscore the importance of (i) stricter grounding constraints for scene-graph generation, and (ii) more conservative prompt-refinement strategies that limit speculative contextual expansion.

## D   RUNTIME AND RESOURCE CHARACTERIZATION

**Setup.** We report end-to-end latency and memory under the following conditions: GPU: `<NVIDIA-A100-SXM4, 80 GB>`, CPU: `<Intel Xeon>`, RAM: `<100 GB>`, framework versions: `<PyTorch 2.5, CUDA 12.1>`, precision: `fp32`, diffusion backbone: `SDEdit-1.5/SDEdit-3.5`, image resolution: $512\times512$, batch size: 1. We denote the number of correction loops by $T = 5$, the denoising steps per edit by NFE $= 50$, and the classifier-free guidance scale by $\gamma = 7$.

Verifier: OWL-ViT `google/owlvit-base-patch32`,
CLIP `openai/clip-vit-large-patch14`,
VLM `meta-llama/Llama-4-Maverick-17B-128E-Instruct`,
LLM `meta-llama/Llama-4-Maverick-17B-128E-Instruct` (used in text-only mode).

Per-iteration times: $t_{\text{SG}}$=3 s, $t_{\text{OWL}}$=300 ms, $t_{\text{CLIP}}$=150 ms, $t_{\text{LLM}}$=2.5 s End-to-end: $t_{\text{total}} \approx 36$ s, peak VRAM: 67 GB.

**Additional Runtime Characterization.** To better contextualize ARTIE's computational footprint, we now make explicit the fixed overhead introduced by the agentic controller per iteration. Aggregating the verification and planning components yields a stable per-iteration overhead of

$$t_{\text{overhead}} = t_{\text{SG}} + t_{\text{OWL}} + t_{\text{CLIP}} + t_{\text{LLM}} \approx 6 \text{ s}$$

This cost remains constant across generative backbones, making ARTIE lightweight relative to the dominant bottleneck: diffusion-based sampling.

**Total Iteration Time Across Backbones.** Table 3 summarizes the total editing time per iteration for several editing backbones.

| Image Generator | $t_{\text{IG}}$ | $t_{\text{total}}$ |
|---|---|---|
| SD-3.5 | 30 s | 36 s |
| InstructPix2Pix | 6 s | 12 s |
| InstructDiffusion | 7 s | 13 s |
| FLUX-Kontext | 28 s | 34 s |

Table 3: Total time per correction iteration, decomposed into generator cost ($t_{\text{IG}}$) and ARTIE's fixed overhead.

**Discussion.** We agree that latency is an important consideration for practical deployment. The updated analysis emphasizes that the overwhelming share of end-to-end runtime arises from the diffusion backbone rather than ARTIE itself:

- **Diffusion dominates computation.** For SD-1.5/3.5 at 50 NFE, $t_{\text{SD}} \approx 30$ s accounts for roughly **83%** of total iteration time. This reflects an intrinsic limitation of high-resolution diffusion sampling on a single A100 GPU.

- **ARTIE introduces only a small, stable overhead.** Its verifier–planner stack contributes $\approx$6 s per iteration, independent of the underlying model, resolution, or instruction complexity.

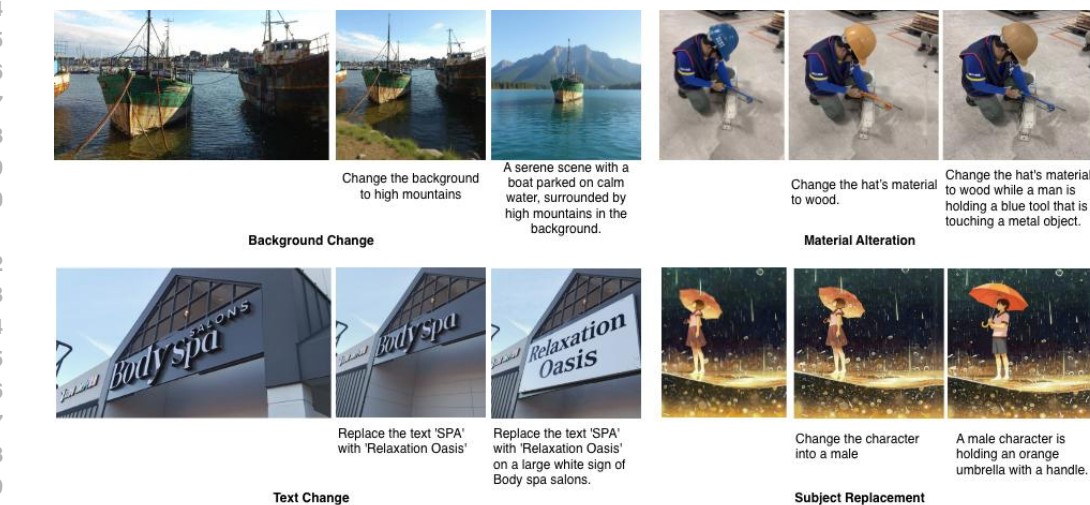

Figure 13: Across four separate edit categories, we show result for FLUX-Kontext and `ARTIE` +FLUX-Kontext along with the prompt and the source image.

- **Fast editors benefit proportionally.** For editing-specialized models with substantially quicker forward passes (e.g., InstructPix2Pix), the diffusion step becomes much lighter while ARTIE's overhead remains constant. Thus, ARTIE inherits the speed advantages of such architectures.

- **Model-agnostic design.** ARTIE does not modify, accelerate, or replace the image generator; it serves purely as an inference-time controller. Consequently, its end-to-end latency is tightly coupled to the backbone's efficiency and cannot overcome the intrinsic sampling cost of slower diffusion models.

## D.1 COMPLEXITY SKETCH

Under fixed verifier costs, the dominant term is diffusion sampling. With image area $A=H\times W$, the per-edit sampling cost scales approximately linearly in NFE and sublinearly to linearly in $A$ depending on attention implementations:

$$t_{\mathrm{SD}} \approx \mathrm{NFE} \cdot \left(\alpha \cdot A + \beta \cdot A \log A + \delta\right), \quad t_{\mathrm{total}} \approx \sum_{i=1}^{T} \left(t_{\mathrm{SG}}^{(i)} + t_{\mathrm{OWL}}^{(i)} + t_{\mathrm{CLIP}}^{(i)} + t_{\mathrm{LLM}}^{(i)}\right) + T \cdot t_{\mathrm{SD}}.$$

Here $\alpha, \beta, \delta$ summarize hardware- and kernel-dependent constants. Total efficiency in this setting is $t_{SD}/t_{total} = 30/36 = 0.83$.

## E QUALITATIVE EXAMPLES FROM GEDIT DATASET

Figure 13 provides clear qualitative examples on GEdit showing how ARTIE resolves the under-editing and over-editing failures observed in the images generated by one-shot editor FLUX-Kontext. The figure includes challenging instances across four GEdit categories. For each case, it shows: (i) the input and instruction, (ii) FLUX-Kontext's output, and (iii) ARTIE + FLUX-Kontext's output with the corresponding refined prompt.

**Mechanism.** ARTIE improves editing quality by enriching prompts with explicit change and preservation constraints based on scene-level diagnostics: i) Under-editing is detected via insufficient grounding of the target attribute (OWL-ViT + CLIP). ARTIE's LLM module then adds missing change specifications. ii) Over-editing is detected via scene-graph misalignment, after which ARTIE inserts preservation triplets and uncertainty-weighted negative constraints to protect non-target regions.

This controlled prompt refinement simultaneously improves the accuracy and fidelity of preservation changes.

**Example (Figure 13).** For the instruction "Change the background to high mountains," FLUX-Kontext either weakly alters the background or unintentionally modifies the boat/water. ARTIE identifies the error type and generates a refined prompt such as: "A boat on calm water with high mountains in the background." This prompt encodes both required change and preservation constraints, leading to an edit that is semantically aligned with the instruction. The refined prompts shown in Figure 13 make the refinement loop explicit and auditable.

# F PROMPTS

## F.1 SOURCE SCENE GRAPH PROMPT

```
You are an expert scene graph generator. You will be given an image
and a "Target Description" of a desired edit.

Your task is to generate a **preservation-centric scene graph** from
 the image.

### CRUCIAL RULE:
The scene graph must **only** contain information that should be **
preserved** during the edit. You must **EXCLUDE** any triplets
describing objects or attributes mentioned in the **Target
Description**. The goal is to create a list of facts about the image
 that should remain unchanged.

### Instructions:
1.  **Output Format:** The output must be a JSON array of triplets.
Each triplet is a list of three strings: `[subject, relationship,
object]`.
2.  **Relationship Constraint:** The `relationship` string must
either be exactly `"has"` or start with `"is"` (e.g., `"is parked on
"`, `"is metallic"`, `"is"`).
3.  **Content Focus:** Since the images primarily feature cars, most
 triplets should describe the car and its parts. You may include
other relevant objects.
4.  **Strict Exclusion:** Do not, under any circumstances, include
information targeted for change in the output graph.
5.  **Generate as many triplets as possible**

---
### In-Context Example:

**Image:** [An image of a red BMW car parked on a road next to a
sidewalk.]
**Target Description:** "The headlights are damaged."

**CORRECT OUTPUT:**
[
["car", "has", "wheel"],
["car", "has", "BMW logo"],
["wheel", "is", "metallic"],
["car", "is parked on", "road"],
["road", "is beside", "sidewalk"]
]

**Reasoning for the example:** The triplet `["headlight", "is", "
there"]` is explicitly **excluded** from the output because the
aspect headlight is part of the Target Description and is meant to
be changed, not preserved.

---
### Your Task:

**Target Description:** {user_description}
```

```
    Now, generate the preservation-centric scene graph following all the
     rules above.
```

## F.2  TARGET SCENE GRAPH PROMPT

We provide the below prompt along with the intermediate image to the VLM to convert the source scene graph into the target scene graph via generating programmatic instructions. After the VLM outputs the edit operations (programmatic instructions), the source scene graph is converted to the target scene graph using a rule-based Python program.

```
    You are an AI Programmer specializing in knowledge graph
    transformations. You will be given a "Source Scene Graph," which is
    a list of `[subject, relation, object]` triplets. You will also be
    given a "New Image".

    ### Core Task:

    Your task is to compare the New Image against the Source Scene Graph
     and generate a JSON list of **edit operations** required to
    transform the Source Scene Graph into a Target Scene Graph that
    accurately represents the New Image.

    ### Defined Operations:

    You must only use the following two operations.

    | Operation       | Parameters         | Description |
    | ---------------| ----------------| ----------------|
    | `ADD_TRIPLET`  | `new_triplet` (a list of three strings)  | Adds
    a new triplet that is now true in the New Image.           |
    | `REMOVE_TRIPLET`| `old_triplet` (a list of three strings)  |
    Removes an old triplet that is no longer true in the New Image.  |

    ### Important Rules:

    1.  **Strictly JSON Output:** Your entire output must be a single,
    valid JSON array of operation objects.
    2.  **Modification as Remove+Add:** To modify a triplet, you must
    first `REMOVE_TRIPLET` with the old triplet, and then `ADD_TRIPLET`
    with the new one.
    3.  **Be Precise:** The `old_triplet` in a `REMOVE_TRIPLET`
    operation must be an *exact match* to a triplet in the Source Scene
    Graph.
    4.  **No Redundancy:** If a triplet from the source graph is still
    true in the new image, generate no operations for it.
    5.  **CRUCIAL Node Constraint for ADD_TRIPLET:** In any `ADD_TRIPLET
    ` operation, the `subject` must be entities that already exist in
    the set of nodes from the Source Scene Graph. **You cannot introduce
     new entities as subjects.** For example, if "windshield" is not a
    node in the source graph, you cannot add a triplet like `["
    windshield", "has", "crack"]`.
    6.  **You don't have to show your reasoning. Just give the output in
     the desired format.**
    ---
    ### In-Context Examples:

    **Example 1: Simple Attribute Change**

    **Source Scene Graph:**
    [
    ["car", "is", "blue"],
    ["car", "has", "wheel"]
```

```
    ]

    **New Image:** [An image of the same car, but it is now red.]

    **VLM Output:**

    [
    {
    "op": "REMOVE_TRIPLET",
    "old_triplet": ["car", "is", "blue"]
    },
    {
    "op": "ADD_TRIPLET",
    "new_triplet": ["car", "is", "red"]
    }
    ]

    ---
    **Example 2: Object Deletion**

    **Source Scene Graph:**
    [
    ["car", "has", "side mirror"],
    ["side mirror", "is", "intact"]
    ]

    **New Image:** [An image of the car, but the side mirror is now
    missing.]

    **VLM Output:**

    [
    {
    "op": "REMOVE_TRIPLET",
    "old_triplet": ["car", "has", "side mirror"]
    },
    {
    "op": "REMOVE_TRIPLET",
    "old_triplet": ["side mirror", "is", "intact"]
    }
    ]

    ---
    **Example 3: Object Addition (Damage)**

    **Source Scene Graph:**
    [
    ["windshield", "is", "intact"]
    ]

    **New Image:** [An image of the windshield, which now has a large
    crack in it.]

    **VLM Output:**
    [
    {
    "op": "REMOVE_TRIPLET",
    "old_triplet": ["windshield", "is", "intact"]
    },
    {
    "op": "ADD_TRIPLET",
    "new_triplet": ["windshield", "is", "cracked"]
    },
```

```
    {
    "op": "ADD_TRIPLET",
    "new_triplet": ["windshield", "has", "crack"]
    },
    {
    "op": "ADD_TRIPLET",
    "new_triplet": ["crack", "is", "large"]
    }
    ]

    ---
    """

    disco_sg1 = f"""
    ### Your Task:

    **Source Scene Graph:**
    {src_sg}

    **New Image:**
    [IMAGE IS PROVIDED]

    **VLM Output:**
```

### F.3 ASPECT EXTRACTION PROMPT

```
    You are a highly skilled prompt analyst. Your task is to read a
    detailed description of a target prompt and extract a concise,
    numbered list of all aspects related to **prompt only**.

    ### Instructions:

    1.  Read the entire input description carefully.
    2.  Identify and isolate every statement or phrase that describes
    the event in the prompt.
    4.  Summarize each distinct point of the prompt into a clear,
    complete sentence, yet very short.
    5.  Format your final output as a pythonic list. And **strictly**
    follow this format. See the output below for reference.

    ---
    ### Example:

    **Input Description:**
    "Make the hydrant all white."

    **Output:**
    ['There is a hydrant', 'The hydrant is white.']

    ---
    ### Your Task:

    **Input Description:**
    {user_description}

    Now generate the output. Do not show your reasoning steps. only
    generate the output.
```

### F.4 PROMPT REWRITING INSTRUCTION

```
1404
1405          You are an expert prompt engineer for diffusion models. Your task is
1406           to analyze a failed image edit and generate a new set of prompts to
               fix it. Do not show your reasoning. Only rewrite the prompt.
1407
1408          You will be given:
1409          1. The Original Prompt.
1410          2. A list of Preservation Failures (P): Triplet-based facts that
1411          should have been kept.
               3. A list of Change Failures (Q): Sentences describing changes that
1412          were not made.
1413
1414          Your goal is to generate a JSON object with three keys:
              1. "refined_positive_prompt": Rewrite the original prompt to be more
1415           descriptive and explicit. It must reinforce the preservation
1416          requirements from P and clearly state the desired changes from Q.
1417          2. "negative_prompt": Create a comma-separated list of keywords
              describing the specific errors to avoid.
1418          3. "negative_prompt_mapping": A JSON object mapping each phrase from
1419           your negative_prompt back to the original failure from list P or Q
1420          that inspired it.
1421
1422          --- EXAMPLE ---
1423          Original Prompt: "A detailed photo showing the front-right side of
              the car after a minor accident. The headlight is visibly cracked,
1424          and there is a noticeable scrape on the fender. The car's blue paint
1425           is otherwise glossy and reflective."
1426          Preservation Failures (P): "[['car', 'is', 'blue'], ['car', 'has', '
              side mirror']]"
1427          Change Failures (Q): "['The headlight is damaged and cracked.', '
1428          There is a noticeable scrap on the fender.']"
1429
1430          JSON Output:
1431          {
1432          "refined_positive_prompt": "A front side of a blue car is met with
              an accident. The headlight of the car is damaged. Also, there is a
1433          noticeable crack in the fender.",
1434          "negative_prompt": "green car, yellow car, no side mirror, missing
1435          mirror, intact headlight, fully furnished fender, undamaged fender",
1436          "negative_prompt_mapping": {
1437              "green car": "['car', 'is', 'blue']",
1438              "yellow car": "['car', 'is', 'blue']",
                  "no side mirror": "['car', 'has', 'side mirror']",
1439              "missing mirror": "['car', 'has', 'side mirror']",
1440              "intact headlight": "The headlight is damaged and cracked.",
1441              "fully furnished fender": "There is a noticeable scrap on the
1442          fender.",
                  "undamaged fender": "There is a noticeable scrap on the fender
1443          .",
1444          }
1445          }
1446          --- END EXAMPLE ---
1447          """
1448          s = f"""
              --- YOUR TASK ---
1449          Original Prompt: {original_prompt}
1450          Preservation Failures (P): {p_failures_str}
1451          Change Failures (Q): {q_failures_str}
1452
1453          JSON Output:
```

1454
1455 ## F.5 CAR125 PRE-ACCIDENT DESCRIPTION GENERATION PROMPT
1456

```
1457          I am providing you an accident description that describes the above
              image. Generate a detailed photorealistic caption that describes the
```

```
   original pre-accident undamaged image given the accident image and
   accident description.
   Accident Description: {acc_desc}
```

## G  USE OF LARGE LANGUAGE MODELS

In adherence to ICLR 2026 policy, we disclose our use of Large Language Models (LLMs) during the preparation of this manuscript. LLMs (Gemini 2.5 Pro, ChatGPT) were utilized as a general-purpose writing assistant for paraphrasing and improving the clarity, readability of sentences, and content organization. The core research ideas, experimental design, results, and their interpretation were conceived and formulated entirely by the authors. The LLM's role was strictly limited to language refinement and did not contribute to the scientific ideation or analysis presented in this work. The authors have reviewed all content and take full responsibility for the final manuscript.

