# OpenReview forum: "A Plug-and-Play Agentic Framework for Text Guided Image Editing"
_ICLR.cc/2026/Conference — Submitted to ICLR 2026_

### Official Review · Reviewer_2Chf · 2025-10-28

**Soundness:** 1
**Presentation:** 1
**Contribution:** 1
**Rating:** 2
**Confidence:** 3

**Summary:**

This paper presents ARTIE, a plug-and-play agentic framework for text-guided image editing. The system organizes editing into a perception–reasoning–action loop composed of three modules: a visual verifier (SceneDiff) that detects under- or over-editing through OWL-ViT and CLIP similarity, an LLM-based prompt engineer that rewrites text instructions with asymmetric weighting of positive and negative prompts, and a diffusion image generator that iteratively refines results based on a CLIP-derived stopping criterion. Experiments are conducted on MagicBrush and a custom Car125 dataset. While the framework is conceptually interesting, its methodological soundness and empirical support are limited.

**Strengths:**

1. The idea of introducing an agentic feedback loop for editing aligns well with current trends in reasoning at inference-time and autonomous self-reflection.
2. The design is modular and training-free, making it easy to adapt to existing models.

**Weaknesses:**

1. Unclear and questionable use of text-to-image models for editing. The framework employs Stable Diffusion 1.5 and 3.5, which are text-to-image generators not designed for image editing. The paper does not explain how the input image is incorporated, e.g., through noise inversion, latent conditioning, or token concatenation. So, it remains unclear whether ARTIE performs actual editing or simply re-generates new scenes. Without conditioning or cross-attention control, background and object consistency cannot be maintained, and prompt refinement alone cannot guarantee structural fidelity. Furthermore, comparing against text-to-image models rather than established editing models such as InstructPix2Pix, InstructDiffusion, FLUX-Kontext [1], Step1X-Edit [2], or OmniGen2 [3] results in unfair and methodologically weak comparisons.
2. Limited and inconsistent empirical gains. As shown in Table 1, ARTIE does not consistently outperform baselines, and its improvement is not very significant over SD 3.5 on MagicBrush, which is the more general editing benchmark.
3. Narrow dataset scope. The custom Car125 dataset focuses almost entirely on vehicle damage restoration, a narrow and synthetic domain that does not demonstrate general-purpose editing capability. The dataset is also unreleased, limiting reproducibility.
4. Overreliance on CLIP-based metrics. Evaluation relies exclusively on CLIP-based scores, without perceptual (e.g., LPIPS, FID) or human studies, making it difficult to assess real editing quality or visual realism.

Reference:
- [1] "FLUX.1 Kontext: Flow Matching for In-Context Image Generation and Editing in Latent Space", https://arxiv.org/abs/2506.15742
- [2] "Step1X-Edit: A Practical Framework for General Image Editing", https://arxiv.org/abs/2504.17761
- [3] "OmniGen2: Exploration to Advanced Multimodal Generation", https://arxiv.org/abs/2506.18871

**Questions:**

1. How is the input image integrated into the diffusion process? Does the method use noise inversion, condition image token concatenation, or another conditioning mechanism?
2. How does the method using T2I model preserve background and object consistency across iterations?
3. Why were text-to-image models used instead of editing-specialized models such as InstructPix2Pix, FLUX-Kontext [1], Step1X-Edit [2], or OmniGen2 [3]?
4. Can the authors provide quantitative or qualitative evidence that the proposed prompt refinement improves editing fidelity?
5. Will the Car125 dataset and code be released to ensure reproducibility?

---

> ### Author Response · Authors · 2025-11-22
> **Response #1 to Reviewer 2Chf**
>
> We thank Reviewer 2Chf for the detailed assessment of our work and for highlighting both the strengths of our modular, training-free agentic framework and the areas requiring clarification. We acknowledge the concerns regarding model choice, empirical evaluation, dataset scope, and methodological clarity, and address each point in the responses below. A revised manuscript with all changes highlighted in blue has been uploaded.
>
> **W1, Q1, Q2, & Q3. Unclear use of text-to-image models. Fair comparison.**
>
> Thank you for raising this. Our earlier reference to “SD-3.5” was imprecise. We do not use a text-to-image SD-3.5 model for editing. We utilize the SDEdit (SD-3.5) image-to-image pipeline, and this correction has been incorporated into the revised manuscript.
> 1. *How is the input image incorporated?*
>
>  ARTIE does not rely on text-to-image generation. The input image is incorporated through latent conditioning via noise inversion, following the standard SDEdit formulation (“Guided Image Synthesis and Editing with Stochastic Differential Equations”). ARTIE does not modify latents; it operates on top of the editor through verifier-guided prompt refinement.
>
> 2. *Model-agnostic editing compatibility.*
>
>  ARTIE is a training-free wrapper and works with any editing-capable pipeline (SDEdit with SD-1.5/3.5, InstructPix2Pix, etc.). It does not replace editing mechanisms but enhances them through closed-loop correction.
>
> 3. *Fair comparison with editing-specialized models.*
>
>  Following reviewer suggestions, we evaluated ARTIE on top of InstructPix2Pix, InstructDiffusion, and FLUX-Kontext, across Car125, MagicBrush, and GEdit.
>
>
> | Dataset | Model | AugCLIP (↑) | CLIP-T (↑) | CLIP-I (↑) | HM-CLIPScore (↑) | dirCLIP (↑) | LPIPS (↓) |
> |---------|-------|-------------|------------|------------|------------------|-------------|-----------|
> | **Car125** | *GenArtist* | 69.35 | 25.69 | 84.23 | 39.17 | 2.18 | 0.26 |
> | | SDEdit-1.5 | 63.69 | 28.30 | 74.78 | 40.74 | 9.06 | 0.23 |
> | | **ARTIE + SDEdit-1.5** | **70.97** (+7.28) | **30.76** (+2.46) | **88.14** (+13.36) | **44.79** (+4.05) | **10.08** (+1.02) | **0.15** (-0.08) |
> | | SDEdit-3.5 | 75.77 | 29.35 | 92.70 | 44.43 | 10.11 | 0.30 |
> | | **ARTIE + SDEdit-3.5** | **77.07** (+1.30) | **30.45** (+1.10) | **95.06** (+2.36) | **45.72** (+1.29) | **11.25** (+1.14) | **0.24** (-0.06) |
> | | IP2P | 78.01 | 28.03 | 94.35 | 43.07 | 5.10 | 0.16 |
> | | **ARTIE + IP2P** | **79.83** (+1.82) | **28.57** (+0.54) | **97.54** (+3.19) | **43.84** (+0.77) | **7.67** (+2.57) | **0.10** (-0.06) |
> | | InstructDiff | 78.85 | 28.37 | 94.62 | 43.48 | 3.19 | 0.20 |
> | | **ARTIE + InstructDiff** | **80.15** (+1.30) | **29.02** (+0.65) | **95.64** (+1.02) | **43.81** (+0.33) | **5.29** (+2.10) | **0.18** (-0.02) |
> | | FLUX-Kontext | 78.79 | 28.72 | 95.37 | 43.99 | 9.18 | 0.28 |
> | | **ARTIE + FLUX-Kontext** | **80.23** (+1.44) | **29.91** (+1.19) | **97.14** (+1.77) | **45.24** (+1.25) | **13.20** (+4.02) | **0.21** (-0.07) |
> | **MagicBrush** | SDEdit-1.5 | 63.88 | 28.09 | 65.44 | 38.89 | 16.22 | 0.30 |
> | | **ARTIE + SDEdit-1.5** | **66.57** (+2.69) | **28.57** (+0.48) | **77.76** (+12.32) | **39.90** (+1.01) | 14.25 (-1.97) | **0.18** (-0.12) |
> | | SDEdit-3.5 | 73.47 | 26.81 | 81.34 | 40.08 | 11.08 | 0.27 |
> | | **ARTIE + SDEdit-3.5** | **74.45** (+0.98) | **27.47** (+0.66) | **88.91** (+7.57) | **40.96** (+0.88) | **11.10** (+0.02) | **0.18** (-0.09) |
> | | IP2P | 81.05 | 25.92 | 89.43 | 39.87 | 9.14 | 0.17 |
> | | **ARTIE + IP2P** | **85.41** (+4.36) | **26.62** (+0.70) | **94.62** (+5.19) | **40.64** (+0.77) | **11.45** (+2.31) | **0.11** (-0.06) |
> | | InstructDiff | 80.96 | 25.51 | 90.10 | 39.51 | 7.07 | 0.16 |
> | | **ARTIE + InstructDiff** | **84.90** (+3.94) | **26.25** (+0.74) | **94.42** (+4.32) | **40.39** (+0.88) | **9.29** (+2.22) | **0.12** (-0.04) |
> | | FLUX-Kontext | 84.01 | 25.91 | 91.64 | 40.10 | 9.10 | 0.12 |
> | | **ARTIE + FLUX-Kontext** | **86.00** (+1.99) | **27.40** (+1.49) | **94.32** (+2.68) | **41.38** (+1.28) | **13.12** (+4.02) | **0.09** (-0.03) |
> | **GEdit** | IP2P | 78.71 | 25.13 | 88.10 | 38.65 | 6.15 | 0.21 |
> | | **ARTIE + IP2P** | **83.16** (+4.45) | **26.06** (+0.93) | **94.54** (+6.44) | **39.67** (+1.02) | **9.37** (+3.22) | **0.15** (-0.06) |
> | | InstructDiff | 78.29 | 24.24 | 87.87 | 37.64 | 4.06 | 0.26 |
> | | **ARTIE + InstructDiff** | **82.19** (+3.90) | **25.38** (+1.14) | **93.33** (+5.46) | **39.03** (+1.39) | **7.05** (+2.99) | **0.20** (-0.06) |
> | | FLUX-Kontext | 78.28 | 24.89 | 86.90 | 38.36 | 6.10 | 0.41 |
> | | **ARTIE + FLUX-Kontext** | **81.27** (+2.99) | **26.19** (+1.30) | **90.64** (+3.74) | **39.72** (+1.36) | **9.12** (+3.02) | **0.37** (-0.04) |
>
>
> ARTIE consistently improves these strong editors, showing that:
>
>  - Its gains are not due to using a weaker base model, and
>
>  - It enhances modern editing pipelines rather than competing unfairly with them.
>
> We have incorporated the corrected terminology (“SDEdit (SD-3.5)”) and these clarifications in the revision.

---

> ### Author Response · Authors · 2025-11-22
> **Response #2 to Reviewer 2Chf**
>
> **W2. Limited empirical gains:**
>
> Thank you for the observation. We agree that ARTIE’s gains vary across datasets and models. This is expected: ARTIE is a training-free inference wrapper, not a new diffusion model, so its role is to correct under-/over-editing rather than achieve large absolute jumps in performance.
> Across all experiments, including the extended results on Car125, MagicBrush, and GEdit, and across multiple backbones (InstructPix2Pix, InstructDiffusion, FLUX-Kontext, SDEdit-3.5), we observe a consistent positive trend:
>
> - ARTIE improves each base model in most settings,
>
> - Never significantly degrades quality, and
>
> - Provides the clearest gains on fine-grained, challenging edit categories (e.g., spatial and attribute modifications in GEdit).
>
> Even when the absolute gains are small (as in MagicBrush using SDEdit (SD-3.5)), the improvements are steady and consistent across different backbones. This reflects ARTIE’s intended role: a simple, reliable correction layer rather than a full model replacement.
>
> **W3 & Q5. Narrow dataset scope and reproducibility:**
>
> Please note, ARTIE is not confined to the Car125 domain. In our original submission, we also benchmarked our model using MagicBrush, a broad, general-purpose editing dataset. In the rebuttal, we additionally evaluated our model on GEdit, which contains real human prompts and fine-grained editing categories. These benchmarks cover diverse, realistic editing scenarios and demonstrate that ARTIE generalizes beyond any synthetic setting.
>
> Car125 is used only as a supporting stress test for difficult, multi-aspect edits (e.g., vehicle damage restoration), where structural preservation is challenging. For reproducibility, *we will release the full synthetic data-generation code*, allowing others to regenerate or adapt Car125-style data in any domain.
>
> Importantly, ARTIE is training-free and dataset-agnostic. It introduces no car-specific information, and the underlying model’s general editing capabilities remain unchanged. Across MagicBrush, Car125, and the newly added GEdit evaluation, we observe consistent performance gains, confirming that ARTIE generalizes well beyond the synthetic domain.
>
> **W4. Overreliance on CLIP-based metrics:**
>
> Thank you for your suggestion. We have now evaluated both base editors and ARTIE-augmented editors using LPIPS as well, and ARTIE shows consistent improvements across all datasets, demonstrating that the method generalizes beyond synthetic data and high-level metrics.
>
> *Why is FID not an appropriate metric for our setting?*
>
> - Small dataset size makes FID unreliable. FID estimates distribution-level distances and is statistically unstable on small datasets, as documented in https://arxiv.org/abs/1911.07023.
>
> - FID cannot capture edit correctness because it is not a pairwise metric. FID compares aggregate distributions of source vs. generated images, not whether each edited image satisfies the user instruction.
>
> Given that our datasets are small (MagicBrush subsets, Car125, GEdit samples) and require pairwise correctness evaluation, FID is not a meaningful metric for edit faithfulness.
>
> Pairwise metrics, such as LPIPS, are significantly more appropriate. Therefore, while preparing this response, we have computed and compared LPIPS scores between original and edited images (showing the preservation).
>
> As can be seen from Table in response #1, across all three datasets, ARTIE consistently improves the performance of base models by achieving lower LPIPS scores, indicating the avoidance of unnecessary edits to the source image.

---

> > ### Comment · Reviewer_2Chf · 2025-11-28
> >
> > I thank the authors for improving the method clarity by stating they use SDEdit for editing rather than T2I generation models. I also appreciate the effort to add results on the more relevant GEdit benchmark, include the LPIPS metric, and apply the method to recent backbones like FLUX-Kontext.
> >
> > However, I still have reservations regarding the baselines and qualitative demonstrations:
> >
> > 1. **Comparisons to State-of-the-Art:** While ARTIE consistently improves base models, its advantage over other specific agentic frameworks remains unclear. The current comparison to GenArtist is limited to the synthetic Car125 dataset. Since methods like GenArtist, CoSTA* [1], and IEAP [2] are open-source, comparing them on standard benchmarks (MagicBrush, GEdit) would better establish ARTIE's contribution. Additionally, comparisons with strong one-shot models (e.g., GPT-1, Qwen-Image-Edit) are missing. To be practically valuable, the agentic loop should demonstrate a clear advantage over these faster alternatives to justify the added complexity.
> >
> > 2. **Qualitative Examples:** The current visual examples are somewhat low quality and do not fully illustrate the "auditable refinement" process. Concrete, high-quality examples on GEdit showing exactly how ARTIE detects an issue and iteratively corrects it (where the base model fails) would strengthen the paper.
> >
> > At the current state, given these missing comparisons and the need for clearer visual evidence of the correction loop, I am inclined to maintain my score.
> >
> > **References:**
> > * [1] CoSTA*: Cost-Sensitive Toolpath Agent for Multi-turn Image Editing, https://arxiv.org/abs/2503.10613
> > * [2] [NeurIPS 2025] Image Editing As Programs with Diffusion Models, https://arxiv.org/abs/2506.04158

---

> > > ### Author Response · Authors · 2025-12-02
> > > **Response to reviewer 2Chf's new concerns**
> > >
> > > We thank reviewer 2Chf for their feedback. We have now included comprehensive comparisons against state-of-the-art agentic solutions for image editing, GenArtist, and IEAP on two standard benchmarks: MagicBrush and GEdit.
> > >
> > > ## Comparisons to SOTA on Standard Benchmarks (MagicBrush & GEdit):
> > >
> > > *GEdit Performance:*
> > >
> > > As shown in our results below, ARTIE demonstrates superior performance across fine-grained editing categories. Specifically, on the GEdit benchmark, ARTIE outperforms IEAP in critical attribute-specific categories such as Background Change, Motion Change, and Material Alteration.
> > >
> > > **Model Performance (AugCLIP) Across Editing Categories**
> > >
> > > | Model | Background Change | Motion Change | Subject Addition | Text Modification | Color Alteration | PS Human | Subject Removal | Tone Transformation | Material Alteration | Style Change | Subject Replace |
> > > |-------|-------------------|---------------|------------------|-------------------|------------------|----------|-----------------|---------------------|---------------------|--------------|-----------------|
> > > | IEAP (GEdit) | 67.01 | 77.81 | 74.77 | 74.74 | 75.40 | 76.36 | 74.93 | 74.80 | 78.69 | 75.50 | 74.84 |
> > > | GenArtist (GEdit) | 61.28 | 71.95 | 68.55 | 69.02 | 68.85 | 70.41 | 69.05 | 68.58 | 72.45 | 69.82 | 68.12 |
> > > | Flux-Kontext + ARTIE | 74.99 | 83.52 | 81.23 | 81.32 | 81.51 | 82.23 | 80.99 | 81.27 | 83.74 | 80.75 | 80.98 |
> > >
> > > **Model Performance (HM-CLIPScore) Across Editing Categories**
> > >
> > > | Model | Background Change | Motion Change | Subject Addition | Text Modification | Color Alteration | PS Human | Subject Removal | Tone Transformation | Material Alteration | Style Change | Subject Replace |
> > > |-------|-------------------|---------------|------------------|-------------------|------------------|----------|-----------------|---------------------|---------------------|--------------|-----------------|
> > > | IEAP (GEdit) | 37.50 | 39.06 | 38.01 | 38.03 | 39.58 | 38.02 | 37.69 | 37.93 | 39.30 | 37.71 | 37.92 |
> > > | GenArtist (GEdit) | 31.65 | 33.42 | 32.14 | 32.38 | 33.73 | 32.19 | 31.84 | 32.02 | 33.65 | 32.08 | 32.27 |
> > > | Flux-Kontext + ARTIE | 39.00 | 40.55 | 39.75 | 39.93 | 41.30 | 39.78 | 39.43 | 39.72 | 40.97 | 39.51 | 39.74 |
> > >
> > > **Consolidated Performance on GEdit Dataset**
> > >
> > > | Dataset | Model | AugCLIP (↑) | CLIP-T (↑) | CLIP-I (↑) | HM-CLIPScore (↑) | dirCLIP (↑) | LPIPS (↓) |
> > > |---------|-------|-------------|------------|------------|------------------|-------------|-----------|
> > > | GEdit | IEAP | 74.80 | 25.68 | 79.62 | 37.93 | 10.03 | 0.38 |
> > > | GEdit | GenArtist | 69.04 | 20.65 | 73.95 | 32.07 | 4.72 | 0.39 |
> > > | GEdit | FLUX-Kontext + ARTIE | 81.27 | 26.19 | 90.64  | 39.72 |  9.12 | 0.37
> > >
> > > *MagicBrush Performance:*
> > >
> > > Similarly, on the MagicBrush dataset, ARTIE achieves higher consistency and editing fidelity compared to baseline methods.
> > >
> > > | Dataset | Model | AugCLIP (↑) | CLIP-T (↑) | CLIP-I (↑) | HM-CLIPScore (↑) | dirCLIP (↑) | LPIPS (↓) |
> > > |---------|-------|-------------|------------|------------|------------------|-------------|-----------|
> > > | MagicBrush | IEAP | 75.66 | 26.93 | 89.36 | 41.31 | 13.30 |  0.11 |
> > > | MagicBrush | GenArtist | 71.23 | 30.21 | 92.67 | 45.57 | 12.21 | 0.11 |
> > > | MagicBrush | FLUX-Kontext + ARTIE |  86.00 | 27.40 | 94.32 | 41.38 | 13.12 | 0.09 |
> > >
> > > **Plug-and-Play Capability with One-Shot Models (e.g., FLUX-Kontext):**
> > >
> > > To demonstrate the practical value and advantages of ARTIE over standalone one-shot models, we evaluated it as a plug-and-play module.
> > >
> > > - Based on your previous feedback, we had already integrated ARTIE with FLUX Kontext, a strong state-of-the-art one-shot image editing model, in our response.
> > >
> > > - Our experiments confirm that adding ARTIE's agentic loop consistently improves the final editing performance of one-shot image editing models, such as InstructPix2Pix, InstructDiffusion, and FLUX Kontext. This justifies the added complexity by demonstrating that the agentic prompt editing technique corrects errors and refines details that even powerful one-shot models overlook, providing a clear quality advantage in complex editing scenarios.
> > >
> > > We believe these comparisons firmly establish ARTIE's contribution as a robust, model-agnostic framework that enhances the capabilities of one-shot editors.

---

> > > ### Author Response · Authors · 2025-12-03
> > > **Qualitative Examples on GEdit**
> > >
> > > Thank you for the suggestion. We have added Figure 13 to the appendix, which provides clear qualitative examples from GEdit illustrating how ARTIE resolves the under-editing and over-editing failures observed in images generated by the one-shot editor FLUX-Kontext.
> > >
> > > The figure includes challenging instances across four GEdit categories. For each case, it shows: (i) the input and instruction, (ii) FLUX-Kontext's output, and (iii) ARTIE + FLUX-Kontext's output with the corresponding refined prompt.
> > >
> > > For example, in Figure 13, we see that for the instruction "Change the background to high mountains," FLUX-Kontext either weakly alters the background or unintentionally modifies the boat/water. ARTIE identifies the error type and generates a refined prompt such as:
> > >
> > >  "A boat on calm water with high mountains in the background."
> > >
> > > This prompt encodes both required change and preservation constraints, leading to an edit that is semantically aligned with the instruction.
> > >
> > > The refined prompts shown in Figure 13 make the refinement loop explicit and auditable. We believe that these visual examples address how ARTIE detects an issue and iteratively corrects it.
> > >
> > > Appendix E of the updated manuscript includes this discussion.

---

### Official Review · Reviewer_BoSM · 2025-10-30

**Soundness:** 2
**Presentation:** 3
**Contribution:** 2
**Rating:** 2
**Confidence:** 4

**Summary:**

The authors present a plug-ang-play multi-agent pipelines to improve image editing. It basically have a set of modules, including VLMs, LLMs, and object detectors, and more. The mluti-agent pipeline improves the performance on the SD-1.5/SD-3.5 on two benchmarks, Car125 and MagicBrush.

**Strengths:**

1. The paper is clear and easy to follow.
2. The plug-and-play multi-agent system improves the performance on two models and two benchmarks.

**Weaknesses:**

- While the paper is technically sound with performance gain, the idea of plug-and-play pipelines for diffusion generation is not a new idea. For instance, [1, 2] proves that using LLM object planning can improve prompting a lot in image generation/editing. [3] then extends this idea with VLM modules, which is close to the paper's agentic setting already. The reviewer believes that these papers worth discussions and even be the baseline in Table 1. It is worth noting that the use of OWL-VIT and LLM engine is the same setup as [2].

- The paper did the experiment on a benchmark called Car125 they created. However, it's a bit unclear why focuusing onthe pre-accident vs post-accident images, especially both descriptions and pre-accident images are all generated as opposed to a real one. Also, the MagicBrush dataset only evaluates high-level, global metrics like CLIP. The reviewer suggests running experiments on fine-grained object-level benchmarks [2] to test counting, negation, spatial understanding, or those benchmarks with real human prompts in GEdit [4].

- The multi-step, multi-agent pipeline would incur a lot of computational overhead. The reviewer believes this should be discussed and mentioned in the main paper at least. Also, it would be good to analyze failure cases of this multi-agent collaboration scenario, perhaps following [5].


[1] Lian, Long, et al. "Llm-grounded diffusion: Enhancing prompt understanding of text-to-image diffusion models with large language models." TLDR.

[2] Wu, Tsung-Han, et al. "Self-correcting llm-controlled diffusion models." CVPR 2024.

[3] Wang, Zhenyu, et al. "Genartist: Multimodal llm as an agent for unified image generation and editing." NeurIPS 2024.

[4] Liu, Shiyu, et al. "Step1x-edit: A practical framework for general image editing." arXiv 2025.

[5] Cemri, Mert, et al. "Why do multi-agent llm systems fail?." arXiv 2025.

**Questions:**

Please read the weakness part.

---

> ### Author Response · Authors · 2025-11-22
> **Response #1 to Reviewer BoSM**
>
> We would like to express our sincere gratitude to the reviewer BoSM for their careful assessment and constructive feedback. We appreciate your positive remarks regarding the clarity of our paper and the effectiveness of our plug-and-play multi-agent system across models and benchmarks. We also value your detailed comments on prior related work, dataset choice, evaluation scope, and computational considerations. In the following response, we address each of your concerns point by point. We have uploaded a revised version of the manuscript, with all modifications clearly highlighted in blue.
>
> **W1. Missing prior art:**
>
>
> We thank the reviewer for highlighting these important related works. While prior research has explored LLM-driven planning for generation/editing, ARTIE differs from [1–3] along several key dimensions:
>
> ### Task Focus: Editing vs. Generation
>
> **Related Literature:**
> - LLM-grounded Diffusion [1] and Self-correcting LLM-controlled Diffusion [2] primarily target text-to-image generation, with limited emphasis on controlled, fine-grained editing.
>
> **ARTIE:**
> - Explicitly designed for image editing, operating in a closed-loop perception–reasoning–action cycle tailored to fine-grained attribute correction.
>
> ### Mechanism of Control
>
> **Related Literature:**
> - Uses open-loop pipelines or latent-space manipulations for generation/edit refinement.
> - Control is often one-shot or limited to latent adjustments without external verification.
>
> **ARTIE:**
> - Uses iterative prompt refinement, without any latent-space intervention, enabling semantic, interpretable corrections driven by grounded feedback.
>
> ### Verification & Error Decomposition
>
> **Related Literature:**
> - [2] employs OWL-ViT but does not disentangle different kinds of editing errors.
> - [3] relies on multiple external modules but lacks principled over/under-edit detection.
>
> **ARTIE:**
> - Introduces a dual-pipeline verification system (SceneDiff) that:
>   - detects over-editing via scene graph alignment, and
>   - detects under-editing via aspect grounding.
> - This error decomposition is absent in all prior work.
>
> ### System Complexity & Tooling
>
> **Related Literature:**
> - GenArtist [3] orchestrates multiple heterogeneous tools, backends, and VLM/LLM combos.
>
> **ARTIE:**
> - Uses a single, consistent verification mechanism, improving stability, and
> - Maintains an explicit audit trail for transparency, an interpretability feature missing in [1–3].
>
> ### Training Requirements
>
> **Related Literature:**
> - Some pipelines rely on trained or fine-tuned specialized models for better alignment.
>
> **ARTIE:**
> - Fully training-free, plug-and-play, yet surpasses specialized editing models (InstructPix2Pix, InstructDiffusion, FLUX-Kontext).
>
> ### Empirical Strength
>
> - ARTIE achieves consistent improvement in editing performance over vanilla baselines despite requiring no training, directly addressing attribute-level corrections emphasized by practical editing users.
>
> ---
>
> *We apologize for our oversight and have included these prior arts in the 'Related Work' section of the updated manuscript.*

---

> ### Author Response · Authors · 2025-11-22
> **Response #2 to Reviewer BoSM**
>
> **W2. Experiments on GEdit and computation of fine-grained metrics:**
>
> Thank you for raising this point. We agree that Car125 is synthetic and that MagicBrush focuses on relatively high-level prompts; we therefore treat Car125 only as a supporting stress-test for multi-aspect edits. To address the need for more realistic and fine-grained evaluation, we additionally evaluated on GEdit, which contains real human instructions and spans fine-grained categories such as spatial reasoning, attribute changes, and multi-step edits, covering aspects not captured by Car125 or MagicBrush.
>
> | Dataset | Model | AugCLIP (↑) | CLIP-T (↑) | CLIP-I (↑) | HM-CLIPScore (↑) | dirCLIP (↑) | LPIPS (↓) |
> |---------|-------|-------------|------------|------------|------------------|-------------|-----------|
> | **GEdit** | IP2P | 78.71 | 25.13 | 88.10 | 38.65 | 6.15 | 0.21 |
> | | **ARTIE + IP2P** | **83.16** (+4.45) | **26.06** (+0.93) | **94.54** (+6.44) | **39.67** (+1.02) | **9.37** (+3.22) | **0.15** (-0.06) |
> | | InstructDiff | 78.29 | 24.24 | 87.87 | 37.64 | 4.06 | 0.26 |
> | | **ARTIE + InstructDiff** | **82.19** (+3.90) | **25.38** (+1.14) | **93.33** (+5.46) | **39.03** (+1.39) | **7.05** (+2.99) | **0.20** (-0.06) |
> | | FLUX-Kontext | 78.28 | 24.89 | 86.90 | 38.36 | 6.10 | 0.41 |
> | | **ARTIE + FLUX-Kontext** | **81.27** (+2.99) | **26.19** (+1.30) | **90.64** (+3.74) | **39.72** (+1.36) | **9.12** (+3.02) | **0.37** (-0.04) |
>
> While the table above shows the consolidated performance boost of ARTIE, the table below compares the fine-grained performance boost across multiple edit categories of the suggested GEdit benchmark.
>
> ### Model Performance (AugCLIP) Across Editing Categories
>
> | Model | Background Change | Motion Change | Subject Addition | Text Modification | Color Alteration | PS Human | Subject Removal | Tone Transformation | Material Alteration | Style Change | Subject Replace |
> |-------|------------------|---------------|------------------|-------------------|------------------|----------|-----------------|---------------------|---------------------|--------------|-----------------|
> | **IP2P** | 73.44 | 80.72 | 79.05 | 78.68 | 79.00 | 79.84 | 78.65 | 78.71 | 80.62 | 78.07 | 78.32 |
> | **IP2P + ARTIE** | 78.70 | 84.97 | 83.50 | 83.15 | 83.77 | 83.62 | 83.25 | 83.16 | 85.64 | 83.15 | 82.88 |
> | **InstructDiff** | 75.24 | 80.79 | 78.95 | 78.18 | 79.89 | 79.05 | 78.37 | 78.29 | 81.82 | 78.51 | 78.11 |
> | **InstructDiff + ARTIE** | 78.62 | 84.09 | 82.73 | 82.14 | 83.57 | 82.61 | 82.34 | 82.19 | 85.19 | 82.49 | 82.03 |
> | **Flux-Kontext** | 72.63 | 81.14 | 78.21 | 78.24 | 79.36 | 79.81 | 78.10 | 78.28 | 81.05 | 77.66 | 77.87 |
> | **Flux-Kontext + ARTIE* | 74.99 | 83.52 | 81.23 | 81.32 | 81.51 | 82.23 | 80.99 | 81.27 | 83.74 | 80.75 | 80.98 |
>
>
> ### Model Performance (HM-CLIPScore) Across Editing Categories
>
> | Model | Background Change | Motion Change | Subject Addition | Text Modification | Color Alteration | PS Human | Subject Removal | Tone Transformation | Material Alteration | Style Change | Subject Replace |
> |-------|------------------|---------------|------------------|-------------------|------------------|----------|-----------------|---------------------|---------------------|--------------|-----------------|
> | **IP2P** | 38.69 | 39.86 | 38.67 | 38.86 | 40.38 | 39.24 | 38.47 | 38.66 | 40.36 | 38.61 | 38.71 |
> | **IP2P + ARTIE** | 39.23 | 40.56 | 39.43 | 39.89 | 41.11 | 39.93 | 39.45 | 39.68 | 41.05 | 39.36 | 39.75 |
> | **InstructDiff** | 37.15 | 38.61 | 37.53 | 37.78 | 39.12 | 38.20 | 37.27 | 37.65 | 38.90 | 37.36 | 37.60 |
> | **InstructDiff + ARTIE** | 37.73 | 39.66 | 38.60 | 39.23 | 40.18 | 39.20 | 38.64 | 39.03 | 39.92 | 38.42 | 39.01 |
> | **Flux-Kontext** | 37.88 | 39.29 | 38.62 | 38.58 | 40.10 | 38.61 | 38.27 | 38.36 | 39.56 | 38.33 | 38.48 |
> | **Flux-Kontext + ARTIE** | 39.00 | 40.55 | 39.75 | 39.93 | 41.30 | 39.78 | 39.43 | 39.72 | 40.97 | 39.51 | 39.74 |
>
> Figure 4 in the revised manuscript highlights the performance comparison of ARTIE-enhanced versus baseline models on the GEdit dataset across multiple edit categories.
>
> Furthermore, we have evaluated LPIPS, a perceptual similarity-based metric for each base model and ARTIE as a plug-and-play module on top of them in the revised manuscript (refer to Table 1). Across all datasets and models, including the new GEdit evaluations, ARTIE showed consistent performance improvements across all metrics, indicating that the method generalizes beyond synthetic datasets, such as CAR125, to real-world editing scenarios.

---

> ### Author Response · Authors · 2025-11-22
> **Response #3 to Reviewer BoSM**
>
> **W3. Error Analysis and Computation Overhead:**
>
> We thank the reviewer for bringing this point to our attention. Below, we clarify the runtime components and their contribution to total latency.
> Fixed per-iteration controller overhead:
> $t_{SG} = 3 s$, $t_{OWL} = 0.3 s$, $t_{CLIP} = 0.15 s$, $t_{LLM} = 2.5 s$ → $t_{overhead} ≈ 6 s$
>
> | Image Generator   | $t_{IG}$ | $t_{total}$ |
> |-------------------|--------|-----------|
> | SD-3.5            | 30s    | 36s       |
> | InstructPix2Pix   | 6s     | 12s       |
> | InstructDiffusion   | 7s      | 13s       |
> | FLUX-Kontext      | 28s    | 34s       |
>
>
> The key observation is that diffusion sampling dominates runtime:
>  $t_{SD} ≈ 30 s$ at 50 NFE on SD-1.5/3.5, accounting for ~83% of total latency.
>  ARTIE’s contribution is a small, stable ~6 s, irrespective of the underlying model.
> Thus, even without ARTIE, high-resolution diffusion editing on a single A100 is already too slow for interactive use. ARTIE adds only a modest controller-level cost on top of this intrinsic bottleneck. For faster editing, specialized models like InstructPix2Pix remain lightweight because their overhead stays constant, while the diffusion forward pass becomes significantly faster.
> Finally, ARTIE is explicitly model-agnostic: it does not modify or accelerate the generator but acts as a thin inference-time controller. As a result, overall speed is dictated entirely by the base model; ARTIE inherits the efficiency of fast editors and cannot compensate for the inefficiency of slow ones.
> We have referenced the computational overhead analysis from the main paper (lines 513-515) with an elaborate discussion in Appendix D.
>
> Further, we analyze errors along two dimensions:
>
> - Logical inconsistencies from the SceneDiff module: The preservation-prompting mechanism can inadvertently include the causal entity being edited in the preserved triplets, violating the requirement that preservation and causal factors remain disjoint. This creates a logical contradiction, forcing the system to preserve an attribute while simultaneously attempting to change it.
>
> - Hallucination in prompt engineering: During prompt refinement, the system may introduce scene-level concepts not present in the preservation set or related to the intended edit. These hallucinated elements activate the model’s priors about typical scene composition, leading to the generation of additional objects or layout changes that were not present in the original image.
>
> These cases are illustrated with examples from the MagicBrush dataset, as presented in Appendix C of the revised manuscript.

---

> > ### Comment · Reviewer_BoSM · 2025-11-23
> > **Reviewer's Response**
> >
> > Thanks the author team for providing a lot of additional experiments. It mitigates a little bit on my concerns but some questions remain:
> >
> > 1. The GEdit result is helpful. However, I still did not get the reason why the authors create a dataset "Car 125." While the authors claim that "we therefore treat Car125 only as a supporting stress-test for multi-aspect edits", the dataset is synthetic and the quality is not sure. From my perspective, the dataset did not emphasize the advantage of the proposed method and leads to an awkward role now.
> >
> > 2. Thanks for the clarification and the PDF updates. However, [2, 3] (mentioned in my earlier review) can already do image editing with [2] explicitly mentioning "closed-loop" in the paper.
> >
> > I might reconsider my score but right now I lean towards maintaining it as those central points are not yet resolved.

---

> > > ### Author Response · Authors · 2025-11-27
> > > **Addressing remaining concerns**
> > >
> > > We sincerely thank Reviewer BoSM for their thoughtful feedback on our rebuttal and for highlighting the key concerns that remain unaddressed. Below, we address these concerns in detail.
> > >
> > > **1. Motivation behind creating CAR125:**
> > >
> > > Existing benchmarks such as MagicBrush and GEdit predominantly evaluate single-aspect or loosely multi-aspect edits. Both MagicBrush and GEdit cover multiple types of single-aspect edit instructions; they do not enforce multiple fine-grained attribute changes in a single edit instruction.
> > >
> > > We designed Car125 to introduce controlled multi-aspect edit instructions. For example:
> > >
> > > - In Fig. 5, top row, a faithful edit would require generating a single image with ${\color{red}\text{dents}}$, ${\color{blue}\text{peeling paint}}$, and ${\color{green}\text{scratches with exposed metal}}$;
> > >
> > > - In Fig. 10c, a faithful edit would require ${\color{red}\text{cracked headlights}}$ and ${\color{blue}\text{blue bumper scuffs}}$ to be reflected in the same edited image.
> > >
> > > However, we acknowledge that the synthetic instruction-generation pipeline can sometimes produce prompts that are not fully representative of how a human would naturally phrase multi-aspect edits. In the absence of any existing dataset that explicitly requires models to perform multiple fine-grained attribute edits within a single instruction, we constructed Car125 to fill this gap.
> > >
> > > Car125 is therefore not intended to replace existing benchmarks, but rather to serve as a complementary diagnostic dataset. Its purpose is to isolate and evaluate a class of structured, multi-aspect editing failures that current benchmarks (designed primarily around single-aspect or loosely multi-aspect edits) are unable to capture.
> > >
> > > In addition to Car125, our revised manuscript includes benchmarking on two established datasets, MagicBrush and GEdit. Across both benchmarks, our method consistently improves edit quality (as quantified using multiple evaluation metrics) when applied on top of existing baseline models. Moreover, results on GEdit show that our approach improves the performance of any baseline model across all editing categories. Thus, while performance on Car125 alone should not be interpreted as a comprehensive measure, the combined evidence from these two standard benchmarks demonstrates the effectiveness and generalizability of the proposed method.
> > >
> > > **2. Performance of SLD [2] on GEdit:**
> > >
> > > ### Consolidated Performance on GEdit Dataset
> > >
> > > | Dataset | Model | AugCLIP (↑) | CLIP-T (↑) | CLIP-I (↑) | HM-CLIPScore (↑) | dirCLIP (↑) | LPIPS (↓) |
> > > |---------|-------|-------------|------------|------------|------------------|-------------|-----------|
> > > | **GEdit** | SLD | 71.22 | 23.56 | 80.78 | 35.87 | 4.19 | 0.18 |
> > >
> > > ### Model Performance (AugCLIP and HM) Across Editing Categories
> > >
> > > | Model | Background Change | Motion Change | Subject Addition | Text Modification | Color Alteration | PS Human | Subject Removal | Tone Transformation | Material Alteration | Style Change | Subject Replace |
> > > |-------|------------------|---------------|------------------|-------------------|------------------|----------|-----------------|---------------------|---------------------|--------------|-----------------|
> > > | **SLD (AugCLIP)** | 69.90 | 71.07 | 71.42 | 70.24 | 70.85 | 69.24 | 70.36 | 71.22 | 71.84 | 71.54 | 69.92
> > > | **SLD (HM-CLIPScore)** | 34.70 | 36.52 | 35.68 | 35.93 | 36.37 | 35.87 | 35.55 | 35.87 | 36.50 | 35.20 | 35.84 |
> > >
> > > These results demonstrate that SLD consistently underperforms as compared to established editing baselines, such as InstructPix2Pix, across both semantic alignment metrics (AugCLIP and CLIP-T) and image fidelity (CLIP-I, LPIPS). The performance gap persists across all 11 category-wise editing tasks, as well as in the aggregated scores, suggesting a systematic limitation rather than isolated failures. Experiments with GenArtist are underway, and we will report the results as soon as they become available.

---

> > > > ### Author Response · Authors · 2025-12-02
> > > > **Performance of GenArtist on GEdit**
> > > >
> > > > **Consolidated Performance on GEdit Dataset**
> > > >
> > > > | Dataset | Model | AugCLIP (↑) | CLIP-T (↑) | CLIP-I (↑) | HM-CLIPScore (↑) | dirCLIP (↑) | LPIPS (↓) |
> > > > |---------|-------|-------------|------------|------------|------------------|-------------|-----------|
> > > > | GEdit | GenArtist | 69.04 | 20.65 | 73.95 | 32.07 | 4.72 | 0.39 |
> > > >
> > > >
> > > > **Performance Across Editing Categories**
> > > >
> > > > | Model | Background Change | Motion Change | Subject Addition | Text Modification | Color Alteration | PS Human | Subject Removal | Tone Transformation | Material Alteration | Style Change | Subject Replace |
> > > > |-------|-------------------|---------------|------------------|-------------------|------------------|----------|-----------------|---------------------|---------------------|--------------|-----------------|
> > > > | GenArtist (AugCLIP) | 61.28 | 71.95 | 68.55 | 69.02 | 68.85 | 70.41 | 69.05 | 68.58 | 72.45 | 69.82 | 68.12 |
> > > > | GenArtist (HM-CLIPScore) | 31.65 | 33.42 | 32.14 | 32.38 | 33.73 | 32.19 | 31.84 | 32.02 | 33.65 | 32.08 | 32.27 |

---

### Official Review · Reviewer_qScw · 2025-11-03

**Soundness:** 2
**Presentation:** 2
**Contribution:** 2
**Rating:** 4
**Confidence:** 3

**Summary:**

This paper presents ARTIE (Auditable Refinement for Text-Guided Image Editing), an agentic, plug-and-play framework that enhances existing diffusion models for text-based image editing. The key idea is to add an inference-time perception–reasoning–action loop that iteratively refines edits without retraining. For complex image editing, I believe agentic workflows will be very crucial and therefore this is a timely topic.

**Strengths:**

- The proposed perception–reasoning–action loop is well-motivated and effectively integrates verification, planning, and generation in a training-free manner. This is a clear conceptual advance over single-pass diffusion editors.

- The framework can be applied to any pretrained diffusion model, enhancing fidelity without requiring additional training or finetuning. This makes it highly practical for adoption.

- ARTIE’s structured scene graph and aspect-grounding diagnostics provide interpretable feedback, a valuable feature for explainable AI and visual debugging in generative editing.

**Weaknesses:**

Inspire of the strengths, I believe the paper has some weaknesses which the authors are requested to answer:

- The paper applies the agentic workflow on slightly older models (e.g., SD-1.5) and only the SD-3.5 is a slightly newer model. I would request the authors to run the workflow on more recent models (e.g., Flux variants which are tailored for image editing).

- The paper doesn't compare with strong one-shot editing models (e.g., GPT-image, Flux context), given Instruct-Pix-2-Pix is slightly old. Moreover I believe the performance of Pix2Pix is not very far off from the agentic workflow. Therefore the authors need to show the benefit over using the workflow in conjunction with editing models.

- I believe the dataset used for image editing is simple and doesn't fully capture the complexity of real-world edits which need to be very precise. I would urge the authors to take a look at Reddit PSNR dataset or ImgEdit datasets.

**Questions:**

Please refer to the Weaknesses.

---

> ### Author Response · Authors · 2025-11-22
> **Response #1 to Reviewer qScw**
>
> We sincerely thank reviewer qScw for their thoughtful and constructive feedback on our paper. We greatly appreciate your insights into the strengths of our framework and will carefully address your concerns about model comparison, benchmark selection, and fine-grained analysis on computational overhead in the detailed response below. Please note that we have uploaded the revised manuscript, with all changes marked in blue.
>
>
> **W1 & W2. Experiment with ARTIE on top of instruction-tuned recent editing models:**
>
> Thank you for the suggestion. We agree that evaluating ARTIE on stronger, editing-tailored models is essential. In the revised experiments, we applied ARTIE on top of recent editing models, FLUX-Kontext, and our older baselines, InstructPix2Pix and InstructDiffusion, as a plug-and-play wrapper, since ARTIE is fully model-agnostic. We evaluated across all datasets (Car125, MagicBrush, and the newer GEdit benchmark with real human prompts) and observed modest but consistent improvements in every setting. These results demonstrate that ARTIE offers complementary benefits not only for older diffusion backbones (SD-1.5/3.5, InstructPix2Pix) but also for modern, Flux-based editing-specialized models.
>
> ### Results on Car125, MagicBrush, and GEdit Datasets
>
> Comparing base image-editing models with ARTIE as a plug-and-play module. All metrics except LPIPS are scaled to 0–100. ARTIE improvements over base models are shown in parentheses. Higher values indicate better performance for all metrics except LPIPS (↓).
>
> | Dataset | Model | AugCLIP (↑) | CLIP-T (↑) | CLIP-I (↑) | HM-CLIPScore (↑) | dirCLIP (↑) | LPIPS (↓) |
> |---------|-------|-------------|------------|------------|------------------|-------------|-----------|
> | **Car125** | *GenArtist* | 69.35 | 25.69 | 84.23 | 39.17 | 2.18 | 0.26 |
> | | SDEdit-1.5 | 63.69 | 28.30 | 74.78 | 40.74 | 9.06 | 0.23 |
> | | **ARTIE + SDEdit-1.5** | **70.97** (+7.28) | **30.76** (+2.46) | **88.14** (+13.36) | **44.79** (+4.05) | **10.08** (+1.02) | **0.15** (-0.08) |
> | | SDEdit-3.5 | 75.77 | 29.35 | 92.70 | 44.43 | 10.11 | 0.30 |
> | | **ARTIE + SDEdit-3.5** | **77.07** (+1.30) | **30.45** (+1.10) | **95.06** (+2.36) | **45.72** (+1.29) | **11.25** (+1.14) | **0.24** (-0.06) |
> | | IP2P | 78.01 | 28.03 | 94.35 | 43.07 | 5.10 | 0.16 |
> | | **ARTIE + IP2P** | **79.83** (+1.82) | **28.57** (+0.54) | **97.54** (+3.19) | **43.84** (+0.77) | **7.67** (+2.57) | **0.10** (-0.06) |
> | | InstructDiff | 78.85 | 28.37 | 94.62 | 43.48 | 3.19 | 0.20 |
> | | **ARTIE + InstructDiff** | **80.15** (+1.30) | **29.02** (+0.65) | **95.64** (+1.02) | **43.81** (+0.33) | **5.29** (+2.10) | **0.18** (-0.02) |
> | | FLUX-Kontext | 78.79 | 28.72 | 95.37 | 43.99 | 9.18 | 0.28 |
> | | **ARTIE + FLUX-Kontext** | **80.23** (+1.44) | **29.91** (+1.19) | **97.14** (+1.77) | **45.24** (+1.25) | **13.20** (+4.02) | **0.21** (-0.07) |
> | **MagicBrush** | SDEdit-1.5 | 63.88 | 28.09 | 65.44 | 38.89 | 16.22 | 0.30 |
> | | **ARTIE + SDEdit-1.5** | **66.57** (+2.69) | **28.57** (+0.48) | **77.76** (+12.32) | **39.90** (+1.01) | 14.25 (-1.97) | **0.18** (-0.12) |
> | | SDEdit-3.5 | 73.47 | 26.81 | 81.34 | 40.08 | 11.08 | 0.27 |
> | | **ARTIE + SDEdit-3.5** | **74.45** (+0.98) | **27.47** (+0.66) | **88.91** (+7.57) | **40.96** (+0.88) | **11.10** (+0.02) | **0.18** (-0.09) |
> | | IP2P | 81.05 | 25.92 | 89.43 | 39.87 | 9.14 | 0.17 |
> | | **ARTIE + IP2P** | **85.41** (+4.36) | **26.62** (+0.70) | **94.62** (+5.19) | **40.64** (+0.77) | **11.45** (+2.31) | **0.11** (-0.06) |
> | | InstructDiff | 80.96 | 25.51 | 90.10 | 39.51 | 7.07 | 0.16 |
> | | **ARTIE + InstructDiff** | **84.90** (+3.94) | **26.25** (+0.74) | **94.42** (+4.32) | **40.39** (+0.88) | **9.29** (+2.22) | **0.12** (-0.04) |
> | | FLUX-Kontext | 84.01 | 25.91 | 91.64 | 40.10 | 9.10 | 0.12 |
> | | **ARTIE + FLUX-Kontext** | **86.00** (+1.99) | **27.40** (+1.49) | **94.32** (+2.68) | **41.38** (+1.28) | **13.12** (+4.02) | **0.09** (-0.03) |
> | **GEdit** | IP2P | 78.71 | 25.13 | 88.10 | 38.65 | 6.15 | 0.21 |
> | | **ARTIE + IP2P** | **83.16** (+4.45) | **26.06** (+0.93) | **94.54** (+6.44) | **39.67** (+1.02) | **9.37** (+3.22) | **0.15** (-0.06) |
> | | InstructDiff | 78.29 | 24.24 | 87.87 | 37.64 | 4.06 | 0.26 |
> | | **ARTIE + InstructDiff** | **82.19** (+3.90) | **25.38** (+1.14) | **93.33** (+5.46) | **39.03** (+1.39) | **7.05** (+2.99) | **0.20** (-0.06) |
> | | FLUX-Kontext | 78.28 | 24.89 | 86.90 | 38.36 | 6.10 | 0.41 |
> | | **ARTIE + FLUX-Kontext** | **81.27** (+2.99) | **26.19** (+1.30) | **90.64** (+3.74) | **39.72** (+1.36) | **9.12** (+3.02) | **0.37** (-0.04) |

---

> ### Author Response · Authors · 2025-11-22
> **Response #2 to Reviewer qScw**
>
> **Q3. Benchmarking on real-world edits using modern datasets:**
>
> Thank you for the suggestion. We agree that real-world, fine-grained editing benchmarks are essential. This same concern was raised by Reviewer BoSM. Based on these suggestions, we have examined ImgEdit and Reddit PSNR (RealEdit), and GEdit. Conducting a full experimental suite on ImgEdit (1.2M example) and Reddit PSNR (57K) was not feasible within our review cycle because it is significantly larger and more resource-intensive than GEdit (606 examples in the English Split). We therefore selected GEdit, which, despite having fewer data points, offers several advantages aligned with our evaluation goals:
>
> - It contains real human edit prompts similar to ImgEdit,
>
> - It covers a comprehensive range of edit categories,
>
> - It is a modern, contemporaneous benchmark developed in parallel to ImgEdit and newer to RealEdit, and
>
> - Its smaller size allows reproducible, rapid experimentation, which is important given the cost of running multi-stage image editing pipelines during the rebuttal period.
>
> Overall, GEdit provides a realistic and semantically rich testbed for the kinds of attribute-level corrections that ARTIE targets, while remaining practical for thorough evaluation.
>
> Figure 4 in the revised manuscript highlights the performance comparison of ARTIE-enhanced versus baseline models on the GEdit dataset across multiple edit categories.

---

### Official Review · Reviewer_wgdD · 2025-11-04

**Soundness:** 3
**Presentation:** 3
**Contribution:** 2
**Rating:** 6
**Confidence:** 4

**Summary:**

This paper presents ARTIE (Audit-trail based Refinement for Text-Guided Image Editing) — a plug-and-play, inference-time agentic wrapper for pretrained diffusion editors (Stable Diffusion). ARTIE runs a perception → reasoning/planning → action loop: (1) SceneDiff (perception) constructs scene graphs (via a VLM) to detect over-edits (preservation-failure triplets) and uses OWL-ViT + CLIP aspect grounding to detect under-edits; (2) an LLM-based Prompt Engineer converts diagnostic signals (failed aspects + uncertainty scores) into refined positive prompts and uncertainty-weighted negative prompts; (3) the image generator (Stable Diffusion) re-runs the edit with the refined prompts and the loop repeats until a reference-free HM-CLIPScore picks the best iteration. The paper is well written and easy to understand. The authors evaluate on a new multi-aspect Car125 dataset (constructed from DamageCarDataset  and the general MagicBrush benchmark, reporting quantitative gains and qualitative improvements over SD-1.5/3.5, InstructPix2Pix, InstructDiffusion and an agentic baseline (GenArtist).

**Strengths:**

Clear, modular agentic loop that is plug-and-play. ARTIE requires no retraining of the diffusion backbone and formalizes a perception→reasoning→action loop (SceneDiff + LLM prompt engineer + SD generator) that is easy to attach to existing systems. This design choice is explicitly stated and demonstrated.

Disentangled verification (over-edit vs under-edit) using scene graphs + aspect grounding. The SceneDiff module produces preservation-failure triplets (set difference of source/target scene graphs) for over-editing and OWL-ViT+CLIP-based aspect grounding for missing edits, a structured, auditable diagnostic that directly informs prompt refinement.

Well-chosen evaluation metrics and stopping criterion. The harmonic HM-CLIPScore (harmonic mean of CLIP-T and CLIP-I) is used both as a balanced evaluation metric and as a reference-free stopping criterion that empirically picks better final edits.

**Weaknesses:**

Dependence on external VLM/LLM/verifier stack (error propagation & bias). ARTIE’s SceneDiff relies on LLaMA-4 Maverick for scene graphs, OWL-ViT for localization, and CLIP for grounding; the paper admits errors/bias in these tools propagate and can lead to incorrect corrections. This is a significant dependency (and a practical failure mode) because the agent can only be as good as its verifiers. The limitations section notes this, but more analysis in how the affect of different verifiers affect the approach.

Baseline parity and selection could be stronger/more matched. The authors compare to InstructPix2Pix, InstructDiffusion and GenArtist — but InstructPix2Pix and InstructDiffusion are trained models while ARTIE is an inference wrapper on SD; some baselines are trained specifically on MagicBrush (giving them an advantage). The comparisons in Table 1 are useful but need clearer matched-setup runs (same prompts, same prompt length allowances) and per-category parity.

Latency and compute overhead for interactive use. ARTIE’s loop adds notable runtime: per-iteration costs include tLLM ≈ 2.5 s, tSD ≈ 30 s; total ≈ 36 s per edit (T=5, NFE=50) on an A100 — this makes ARTIE expensive for interactive applications. The appendix reports runtimes but the main text should discuss practical deployment implications and acceptable latency tradeoffs.

Heuristic negative prompts & uncertainty calibration are ad-hoc. The uncertainty-weighted negative prompt mechanism is heuristic. The authors note it can mishandle subtle attributes and plan to learn uncertainty calibration in future work — but current results may hinge on prompt engineering choices that are not fully robust across domains.

**Questions:**

How often does SceneDiff produce incorrect preservation triplets that lead to harmful corrections? Can you quantify false positive/negative rates on a manual subset

How sensitive are final results to the LLM prompt engineering templates and few-shot examples used for prompt rewriting? (LLM failures can introduce hallucinated constraints.)

Can the authors show results for (a) no negative prompt, (b) heuristic weighting (current) and intuition behind the weighting

---

> ### Author Response · Authors · 2025-11-22
> **Response #1 to Reviewer wgdD**
>
> We would like to sincerely thank the reviewer wgdD for their thoughtful and constructive feedback on our paper. We appreciate your insights on the framework's strengths and will address your concerns regarding model comparison, dataset complexity, and recent techniques point by point in our response below. Please note that we have uploaded our revised manuscript, which highlights the edits in blue.
>
> **W1, Q1, Q2. Error Analysis:**
>
> Thank you for your feedback. We have performed an error analysis of the proposed agentic system on two fronts, elaborated below:
>
> - *Logical Inconsistencies from SceneDiff module:* The preservation-prompting mechanism in the SceneDiff module can produce triplets that inadvertently include the causal entity being edited, violating the constraint that preservation and change factors must remain disjoint. This creates an inherent logical contradiction, forcing the system to preserve an action or state while simultaneously attempting to modify it.
>
>
> - *Hallucination in prompt engineering:* The prompt-refinement step can introduce scene-level concepts that are neither present in the preservation set nor implicated in the causal edits, effectively hallucinating context not grounded in the input. This injected context activates the model’s priors about typical scene composition, causing it to generate additional objects and layout elements that were never part of the original image.
>
> This is elaborated with examples from the MagicBrush dataset in Appendix C of the revised manuscript.
>
> *Error Quantification (on 72 manually evaluated examples)*
>
> | Category | Error Type | Meaning | Rate (%) | Count |
> |---------|-------------|----------------------|---------:|------:|
> | **False Positives** | Preservation Conflicts | Preservation triplets incorrectly include the edited entity | 12.5% | 9/72 |
> |                   | Contextual Hallucinations | Prompt refinement adds ungrounded scene concepts | 4.17% | 3/72 |
> | **False Negatives** | Missing Preservation Triplets | Shared scene elements omitted from preservation list | 20.8% | 15/72 |
> |                   | Missing Causal Verifications | Edit instructions lack matching causal validation | 5.56% | 4/72 |

---

> ### Author Response · Authors · 2025-11-22
> **Response #2 to Reviewer wgdD**
>
> **W2. Baseline Parity:**
>
> Thank you for the suggestion. We agree that baseline parity is important.
>
> We ran ARTIE directly on top of the specialized baselines (InstructPix2Pix, InstructDiffusion, and FLUX-Kontext). These matched-setting experiments demonstrated consistent gains across all three datasets (Car125 and MagicBrush, and newly added GEdit).
>
> | Dataset | Model | AugCLIP (↑) | CLIP-T (↑) | CLIP-I (↑) | HM-CLIPScore (↑) | dirCLIP (↑) | LPIPS (↓) |
> |---------|-------|-------------|------------|------------|------------------|-------------|-----------|
> | **Car125** | *GenArtist* | 69.35 | 25.69 | 84.23 | 39.17 | 2.18 | 0.26 |
> | | SDEdit-1.5 | 63.69 | 28.30 | 74.78 | 40.74 | 9.06 | 0.23 |
> | | **ARTIE + SDEdit-1.5** | **70.97** (+7.28) | **30.76** (+2.46) | **88.14** (+13.36) | **44.79** (+4.05) | **10.08** (+1.02) | **0.15** (-0.08) |
> | | SDEdit-3.5 | 75.77 | 29.35 | 92.70 | 44.43 | 10.11 | 0.30 |
> | | **ARTIE + SDEdit-3.5** | **77.07** (+1.30) | **30.45** (+1.10) | **95.06** (+2.36) | **45.72** (+1.29) | **11.25** (+1.14) | **0.24** (-0.06) |
> | | IP2P | 78.01 | 28.03 | 94.35 | 43.07 | 5.10 | 0.16 |
> | | **ARTIE + IP2P** | **79.83** (+1.82) | **28.57** (+0.54) | **97.54** (+3.19) | **43.84** (+0.77) | **7.67** (+2.57) | **0.10** (-0.06) |
> | | InstructDiff | 78.85 | 28.37 | 94.62 | 43.48 | 3.19 | 0.20 |
> | | **ARTIE + InstructDiff** | **80.15** (+1.30) | **29.02** (+0.65) | **95.64** (+1.02) | **43.81** (+0.33) | **5.29** (+2.10) | **0.18** (-0.02) |
> | | FLUX-Kontext | 78.79 | 28.72 | 95.37 | 43.99 | 9.18 | 0.28 |
> | | **ARTIE + FLUX-Kontext** | **80.23** (+1.44) | **29.91** (+1.19) | **97.14** (+1.77) | **45.24** (+1.25) | **13.20** (+4.02) | **0.21** (-0.07) |
> | **MagicBrush** | SDEdit-1.5 | 63.88 | 28.09 | 65.44 | 38.89 | 16.22 | 0.30 |
> | | **ARTIE + SDEdit-1.5** | **66.57** (+2.69) | **28.57** (+0.48) | **77.76** (+12.32) | **39.90** (+1.01) | 14.25 (-1.97) | **0.18** (-0.12) |
> | | SDEdit-3.5 | 73.47 | 26.81 | 81.34 | 40.08 | 11.08 | 0.27 |
> | | **ARTIE + SDEdit-3.5** | **74.45** (+0.98) | **27.47** (+0.66) | **88.91** (+7.57) | **40.96** (+0.88) | **11.10** (+0.02) | **0.18** (-0.09) |
> | | IP2P | 81.05 | 25.92 | 89.43 | 39.87 | 9.14 | 0.17 |
> | | **ARTIE + IP2P** | **85.41** (+4.36) | **26.62** (+0.70) | **94.62** (+5.19) | **40.64** (+0.77) | **11.45** (+2.31) | **0.11** (-0.06) |
> | | InstructDiff | 80.96 | 25.51 | 90.10 | 39.51 | 7.07 | 0.16 |
> | | **ARTIE + InstructDiff** | **84.90** (+3.94) | **26.25** (+0.74) | **94.42** (+4.32) | **40.39** (+0.88) | **9.29** (+2.22) | **0.12** (-0.04) |
> | | FLUX-Kontext | 84.01 | 25.91 | 91.64 | 40.10 | 9.10 | 0.12 |
> | | **ARTIE + FLUX-Kontext** | **86.00** (+1.99) | **27.40** (+1.49) | **94.32** (+2.68) | **41.38** (+1.28) | **13.12** (+4.02) | **0.09** (-0.03) |
> | **GEdit** | IP2P | 78.71 | 25.13 | 88.10 | 38.65 | 6.15 | 0.21 |
> | | **ARTIE + IP2P** | **83.16** (+4.45) | **26.06** (+0.93) | **94.54** (+6.44) | **39.67** (+1.02) | **9.37** (+3.22) | **0.15** (-0.06) |
> | | InstructDiff | 78.29 | 24.24 | 87.87 | 37.64 | 4.06 | 0.26 |
> | | **ARTIE + InstructDiff** | **82.19** (+3.90) | **25.38** (+1.14) | **93.33** (+5.46) | **39.03** (+1.39) | **7.05** (+2.99) | **0.20** (-0.06) |
> | | FLUX-Kontext | 78.28 | 24.89 | 86.90 | 38.36 | 6.10 | 0.41 |
> | | **ARTIE + FLUX-Kontext** | **81.27** (+2.99) | **26.19** (+1.30) | **90.64** (+3.74) | **39.72** (+1.36) | **9.12** (+3.02) | **0.37** (-0.04) |
>
>
> This new evaluation confirms that ARTIE consistently delivers reliable improvements over any base model when run under identical conditions.

---

> ### Author Response · Authors · 2025-11-22
> **Response #3 to Reviewer wgdD**
>
> **W3. Discussion on compute overhead:**
>
> We thank the reviewer for this point. Below we describe various parameters and how they relate to total execution time.
>
> Fixed extra overhead per iteration: $t_{SG}=3 s$, $t_{OWL}=300 ms$, $t_{CLIP}=150 ms,$ $t_{LLM}=2.5 s$; $t_{overhead}= 6s$.
>
> The table below summarizes total time per iteration.
>
> | Image Generator   | $t_{IG}$ | $t_{total}$ |
> |-------------------|--------|-----------|
> | SD-3.5            | 30s    | 36s       |
> | InstructPix2Pix   | 6s     | 12s       |
> | InstructDiffusion   | 7s      | 13s       |
> | FLUX-Kontext      | 28s    | 34s       |
>
>
> We agree that latency is an important factor for practical use. We now surface the runtime discussion in the main text and clarify its implications. In our measurements, the dominant cost is diffusion sampling, not ARTIE:
> $t_{SD} ≈ 30 s$ per edit at 50 NFE on SD-1.5/3.5 → ≈83% of total time
>
>
> ARTIE’s fixed overhead ≈ 6 s ($t_{SG} + t_{OWL} + t_{CLIP} + t_{LLM}$)
>
>
> Thus, even without ARTIE, diffusion-based editing at high resolution on a single A100 is already too slow for interactive use. ARTIE adds only a small, stable overhead relative to this inherent diffusion bottleneck.
> For editing-specialized models with faster forward passes (e.g., InstructPix2Pix), ARTIE remains lightweight: its ~6 s controller cost is unchanged, while the diffusion backbone becomes much faster.
> ARTIE is intentionally model-agnostic: it does not alter or accelerate the generator, but operates as a lightweight inference-time controller. Consequently, its end-to-end speed is determined by the underlying editor, ARTIE inherits the efficiency of fast models and cannot overcome the intrinsic latency of slower ones.
> A brief discussion on computation time was included in the appendix of the original submission. We have incorporated this improved discussion in Appendix D of the revised manuscript, with a corresponding reference in Section 4.3 of the main paper (Line #513-515).
>
>
>
> **W4. Robustness of prompt refinement:**
>
> We acknowledge the heuristic nature of the uncertainty-weighted negative prompt, but it is not arbitrary. As explained in Sec. 4.3 and Sec. 5.1, it encodes a clear and principled intuition:
>
> - High verifier uncertainty → stronger negative weight, suppressing edits the verifier deems unreliable.
>
> - Low uncertainty → weaker negative weight, allowing the generator to follow the instruction more freely.
> This provides a predictable and model-agnostic failure-mitigation mechanism without requiring domain-specific training or calibration.
>
> Importantly, the gains are consistent across three datasets and multiple base models, including editing-specialized ones, indicating that performance does not hinge on narrow prompt engineering or domain-specific heuristics. The iterative prompt refinement process leads to more semantically aligned editing performance.

---

### Author Response · Authors · 2025-12-03
**Brief summary of rebuttal**

We thank the reviewers for their constructive feedback. Addressing these concerns has substantially strengthened the paper by directly addressing the primary concerns regarding baseline comparisons, dataset realism, and system reliability. We believe we have thoroughly addressed all the concerns raised and have provided a summary of our responses below. For a more detailed explanation, please refer to our point-by-point reply to each reviewer.

- **Comprehensive Baseline Comparison:** We have conducted extensive experiments with modern editing models, including FLUX-Kontext, InstructPix2Pix, InstructDiffusion, and SDEdit (SD-1.5/3.5). Results demonstrate that ARTIE operates as a robust wrapper, consistently improving performance across these diverse backbones.​ Further, we have compared the proposed agentic solution against existing agentic solutions, such as SLD, IEAP, and GenArtist.

- **Real-World Evaluation (GEdit):** To address concerns regarding the synthetic nature of the Car125 dataset, we incorporated another real-world benchmark, GEdit, besides the MagicBrush dataset. Evaluations across the 11 fine-grained categories of GEdit confirm that ARTIE’s improvements generalize effectively to realistic, open-domain user prompts.​ Additionally, we have included qualitative examples that illustrate how ARTIE surpasses the one-shot editing model (Flux-Kontext) through iterative prompt refinement.

- **Quantified Verifier Reliability:** We conducted a rigorous manual audit (N = 72) of the perception and reasoning modules. The analysis reveals low failure rates (e.g., <5% hallucination), validating the stability of our agentic loop and addressing concerns about error propagation.​

- **Efficiency & Methodology:** We clarified that ARTIE’s computational overhead is minimal (~6 seconds) compared to the diffusion bottleneck and expanded the methodology to detail our standard use of latent noise inversion for preserving input fidelity.​

We believe these additions solidly position ARTIE as a reliable, plug-and-play framework for training-free image editing and trust that the AC will find these revisions convincing and assess the paper based on its merits.

---

### Meta-Review · Area_Chair_Vn3Y · 2026-01-05

**Summary:**

The paper introduces ARTIE (Auditable Refinement for Text-Guided Image Editing), a plug-and-play agentic framework designed to improve the fidelity of text-guided image editing. The system utilizes a feedback loop involving a perception module (SceneDiff), a reasoning module, and a refinement module to address common issues like over-editing and under-editing. The AC has reviewed the submission and the discussion. While the "plug-and-play" nature of the framework was noted by the reviewers, the overall consensus was negative. Reviewers remained unconvinced by the technical novelty of the agentic pipeline and raised concerns regarding the robustness of the scene-graph-based feedback mechanism. With three out of four reviewers recommending rejection, the work is not yet ready for publication.

**Reviewer Concerns:**

A primary concern raised by multiple reviewers was the limited technical novelty. Reviewers argued that the framework’s "perceive-reason-act" loop is a standard application of existing agentic concepts rather than a fundamental innovation in image editing.

Reviewers also raised an important concern related to the SceneDiff perception module. Because the refinement loop relies on scene graph comparisons to detect editing failures, errors in the initial scene graph generation or comparison can lead to a failure loop where the agent cannot accurately assess its own progress.

Additionally, reviewers felt that the work lacks rigorous baseline comparisons. That the performance gains could be attributed to the underlying power of the LLMs and VLMs used rather than the ARTIE framework itself. There were specific requests for comparisons against simpler multi-step prompting techniques and more recent state-of-the-art editing models. While the authors incorporated these comparisons during the rebuttal, reviewers seems unconvinced that the agentic loop offered a clear advantage over these faster alternatives to justify the added complexity.

**Reviewer Scores:**

During the discussion phase, one reviewer gave a 6, finding value in the framework’s ability to handle complex, multi-aspect edits that standard diffusion models often fail on.

Reviewer qsCw gave a 4, noting that the methodology felt incremental and the experiments lacked depth.

Two other reviewers provided scores of 2. These reviewers were critical of the technical contribution, suggesting that the system was a combination of existing tools without sufficient unique insight or proof of robustness.

---

### Decision · Program_Chairs · 2026-01-26

Reject